# Thioesterase superfamily member 1 undergoes stimulus-coupled conformational reorganization to regulate metabolism in mice

Yue Li [1,2], Norihiro Imai[3], Hayley T. Nicholls [3], Blaine R. Roberts [4], Samaksh Goyal [1,2], Tibor I. Krisko [3], Lay-Hong Ang [1,2], Matthew C. Tillman[4], Anne M. Roberts[4], Mahnoor Baqai [1,2], Eric A. Ortlund [4], David E. Cohen [3✉] & Susan J. Hagen [1,2✉]

In brown adipose tissue, thermogenesis is suppressed by thioesterase superfamily member 1 (Them1), a long chain fatty acyl-CoA thioesterase. Them1 is highly upregulated by cold ambient temperature, where it reduces fatty acid availability and limits thermogenesis. Here, we show that Them1 regulates metabolism by undergoing conformational changes in response to β-adrenergic stimulation that alter Them1 intracellular distribution. Them1 forms metabolically active puncta near lipid droplets and mitochondria. Upon stimulation, Them1 is phosphorylated at the N-terminus, inhibiting puncta formation and activity and resulting in a diffuse intracellular localization. We show by correlative light and electron microscopy that Them1 puncta are biomolecular condensates that are inhibited by phosphorylation. Thus, Them1 forms intracellular biomolecular condensates that limit fatty acid oxidation and suppress thermogenesis. During a period of energy demand, the condensates are disrupted by phosphorylation to allow for maximal thermogenesis. The stimulus-coupled reorganization of Them1 provides fine-tuning of thermogenesis and energy expenditure.

[1] Division of General Surgery, Department of Surgery, Beth Israel Deaconess Medical Center, Boston, MA, USA. [2] Department of Surgery, Harvard Medical School, Boston, MA, USA. [3] Division of Gastroenterology and Hepatology, Department of Medicine, Weill Cornell Medical College, New York, NY, USA. [4] Department of Biochemistry, Emory University, Atlanta, GA, USA. ✉email: dcohen@med.cornell.edu; shagen@bidmc.harvard.edu

ntracellular triglycerides are stored within lipid droplets (LD) that are juxtaposed to mitochondria in the cytoplasm[1]. This close relationship facilitates the rapid transfer of fatty acids to mitochondria, which are generated by lipolysis in response to cellular energy demands[2]. One such demand is non-shivering thermogenesis, which is mediated by brown adipose tissue (BAT) in response to cold exposure to generate heat[3]. This occurs, at least in part, when mitochondrial β-oxidation is uncoupled from oxidative phosphorylation via uncoupling protein 1. The hydrolysis of LD-derived triglycerides is initiated following the release of norepinephrine from neurons, which binds to β3 adrenergic receptors on brown adipocytes to stimulate a cell signaling cascade that activates adenylyl cyclase to generate cAMP and activate protein kinase A (PKA)[4]. In brown adipocytes, the generation of fatty acids for mitochondrial oxidation requires PKA to phosphorylate/activate perilipin, a regulatory membrane-bound protein that encircles LD. The function of perilipin is to protect triglycerides within the LD from lipolysis, and its phosphorylation by PKA allows access of cytoplasmic adipose triglyceride lipase (ATGL), hormone-sensitive lipase (HSL), and monoacylglycerol lipase (MAGL), to generate free fatty acids through sequential lipolytic steps[5]. Once fatty acids are free in the cytoplasm, they are esterified with coenzyme A (CoA) by long-chain acyl-CoA synthetase 1 (ACSL1) to form fatty acyl-CoAs, which are transported into mitochondria via carnitine palmitoyl transferase 1 and metabolized to produce heat[6]. Fatty acyl-CoA molecules can also be hydrolyzed to fatty acids in the cytoplasm by acyl-CoA thioesterase (Acot) isoforms[7,8]. The fatty acids generated can be utilized to make additional fatty acyl-CoAs that are transported into mitochondria or can be returned to the LD for storage.

Thioesterase superfamily member 1 (Them1), which is also known as brown fat inducible thioesterase or steroidogenic acute regulatory protein-related lipid transfer (START) domain 14 (StarD14)/Acot11, plays an important role in energy homeostasis[9,10]. Mice with the genetic deletion of Them1 exhibit increased energy expenditure, which results in reduced weight gain when challenged with a high fat diet despite high food consumption[10]. This is attributable to increased mitochondrial fatty acid oxidation, and mechanistic studies have demonstrated that Them1 functions to suppress energy expenditure by limiting triglyceride hydrolysis in BAT, thus inhibiting the mitochondrial oxidation of LD-derived fatty acids[10,11].

Them1 is comprised of tandem N-terminal thioesterase domains and a C-terminal lipid-binding steroidogenic acute regulatory-related lipid transfer (START) domain. Proteomics has revealed that there are a number of BAT-selective phosphorylation sites near the N-terminus of Them1 in vivo, most notable are serine residues at (S) 15, 18, and 25[12]. Here, we show experimentally that Them1 is phosphorylated at the N-terminus after stimulation in vitro and exists in two different phosphorylation-dependent conformational states: punctate and diffuse. Punctate refers to small, highly concentrated aggregations of Them1 that localize to regions of the cytoplasm that interface with LD and mitochondria, whereas diffuse refers to homogeneously distributed Them1 across the cell cytoplasm and nucleus. We demonstrate that β-adrenergic stimulation with NE, which is used to mimic cold exposure, activates a signaling cascade that results in Them1 phosphorylation and a diffuse localization. A functional analysis revealed that Them1 in punctate form suppresses fatty acid oxidation, whereas this suppression is abrogated when it is diffusely distributed in the cell cytoplasm. Overall, these findings highlight the importance of Them1 phosphorylation in the regulation of thermogenesis in BAT and lend support for targeting Them1 for the management of obesity-related disorders.

## Results

**N-terminal serine phosphorylation of Them1.** To assess the BAT-selective phosphorylation of Them1, we consulted the phosphomouse database[12], which revealed phosphopeptides containing phosphorylation events at serine (S) 15, 18, and 25 in the N-terminus of Them1 in vivo within BAT, which are highly conserved (Supplementary Fig. 1).

To explore the role of Them1 S-phosphorylation in regulating metabolism, cultured immortalized brown adipose cells (iBAs) from mouse BAT were used as an in vitro model. Because iBAs did not express Them1 mRNA or protein, as was the case for primary cultured brown adipocytes[11], Them1 expression required plasmid or adenoviral transfection (Supplementary Fig. 2a–c) to study its biochemical and physiological characteristics in vitro. To examine whether Them1 is S-phosphorylated after stimulation, we determined the aggregate abundance of phosphopeptides in the N-terminus of Them1 by mass spectrometry after stimulation with phorbol 12-myristate 13-acetate (PMA; Fig. 1a, b) normalized to hormone-sensitive lipase as a housekeeping phosphopeptide

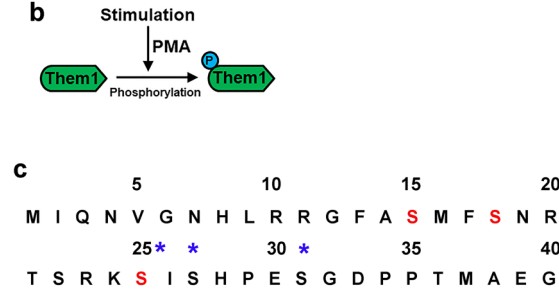

**Fig. 1 Regulation of Them1 phosphorylation and its subcellular localization in iBAs. a** LC-MS/MS data for iBAs expressing Ad-Them1-EGFP stimulated with PMA for 0–4 h. The data are presented as normalized aggregate abundance of N-terminal phosphopeptides using hormore sensitive lipase as a reference for normalization, which does not change after PMA stimulation (see Supplementary Fig. 3). Regression line indicates a positive and significant correlation between phosphorylation events at the N-terminus and time after stimulation. Data are means ± SE for $n = 3$ different experiments/timepoint. Statistical significance was determined by ANOVA on the regression line, where $P < 0.001$. Normality test passed ($P = 0.759$) and constant variance test, via Spearman Rank Correlation, passed ($P = 0.460$). **b** Schematic diagram of the experimental design in **a**. **c** N-terminal amino acid sequence of Them1 showing specific phosphorylation events at S25, S27, and S31 in the Them1 sequence (blue asterisks) as determined by LC-MS/MS. Phosphorylation events in vivo at S15, S18, and S25 are represented by red font.

that does not change with PMA stimulation (Supplementary Fig. 3). This resulted in a linear increase in phosphorylation events over time after stimulation (Fig. 1a), with serine phosphorylation events that could be measured at S25, S27, and S31 (Fig. 1c). Taken together, these in vivo and in vitro results suggested that stimulus-mediated S-phosphorylation within the N-terminus could control Them1 metabolic activity in BAT.

**Phosphorylation-dependent changes in Them1 localization after stimulation**. In transfected cells, Them1 exhibited a punctate intracellular localization (Fig. 2a). When cells co-stained for LD and mitochondria were reconstructed from 3-D stacks, Them1 was closely associated, but did not co-localize, with either of these organelles (Fig. 2b). The punctate localization of Them1 was not due to EGFP aggregation, per se, as evidenced by the diffuse intracellular localization of EGFP in cells transfected with EGFP alone (Supplementary Fig. 4a). Furthermore, Them1 formed puncta in transfected cells that co-localized with Them1-EGFP (Supplementary Fig. 4b), and puncta were also present when Them1 was transfected without the EFGP tag (Supplementary Fig. 4c). The antibody used to detect Them1 was made and characterized by us[11,13] and the newly prepared antibody, which was also affinity purified using a Them1 peptide fragment, was specific for Them1 in BAT and iBAs in immunoblots (Supplementary Fig. 5a) and specifically labeled structures in BAT from wild-type but not in Them1-deficient mice (Supplementary Fig. 5b–d).

We next explored the hypothesis that the stimulus-mediated phosphorylation of Them1 facilitates changes in its localization. When differentiated iBAs were incubated with PMA, which activates protein kinase C (PKC), and the localization of Them1 was examined in fixed or living cells, PMA initiated the dissolution of Them1 from puncta (Fig. 2c, d). The dissolution of puncta resulted in Them1 uniformly distributed within the cell cytoplasm (Fig. 2d), which occurred over a 4-h period after PMA stimulation (Fig. 2e) but not with vehicle alone (Supplementary Fig. 6), as determined by using time-lapse live-cell microscopy. We further observed that the isoform-specific PKCβ inhibitor ruboxistaurin (LY333531 or LY) completely blocked the dissolution of puncta in PMA-treated cells (Fig. 2d).

We next sought to identify upstream signaling events and define the pathway leading to Them1 phosphorylation (Fig. 2c). iBAs incubated with the PKA inhibitor PKI or the ATGL inhibitor atglistatin prior to stimulation with PMA had a diffuse Them1 distribution in the cytoplasm (Fig. 2f). Similarly, iBAs stimulated with forskolin (Fig. 2g), which activates adenyl cyclase and protein kinase A (PKA) upstream of PKC (Fig. 2c), or with norepinephrine, which activates both PKA and PKC (Fig. 2c) resulted in diffusely localized Them1, which was blocked by inhibiting downstream effectors with PKI, Atglistatin, or LY, respectively (Fig. 2g, h). DAG released by ATGL is in the sn-1,3 or sn-2,3 stereo isoform, so that triacylglycerol-derived DAG cannot activate PKC without isomerization to sn-1,2 DAG[14]. This event was not investigated, but would be required to confirm the pathway details (Fig. 2c). Overall, these results suggest that norepinephrine stimulation activates PKA, and the activation of PKC occurs downstream of PKA activation (Fig. 2c). Measuring the volume of intracellular Them1 in iBAs transfected with Them1-EGFP with or without inhibitors (Fig. 2i, schematic; green EFGP fluorescence signal) further demonstrated that Them1 in puncta is concentrated and occupies significantly less volume than occurs when Them1-EGFP is diffusely distributed after stimulation with PMA or forskolin (Fig. 2i). The dissolution of Them1 from puncta after stimulation with PMA or forskolin was

not attributable to the synthesis of new Them1-EGFP protein, because inhibition of protein synthesis using cycloheximide during the 4 h period of stimulation had no effect on Them1 localization or protein concentration (Supplementary Fig. 7).

The canonical PKC binding sequence, R/K-X-S-X-R/K where X at +1 is a hydrophobic residue, is not present within the N-terminal Them1 sequence (Fig. 1c). However, our pathway suggested that PKC was involved in the dissolution of puncta. To examine this experimentally, we used an antibody specific for the canonical PKC binding sequence, which showed that no antibody recognition of PKC-mediated S-phosphorylated Them1 occurred (Supplementary Fig. 8). Instead, at least six proteins in iBAs were time-dependently phosphorylated by PKC after PMA stimulation (Supplementary Fig. 8). These results support that Them1 per se is not a PKC substrate but that PKC may be indirectly involved in the dissolution of puncta after PMA stimulation (Fig. 2c).

We next sought evidence that the stimulus-coupled changes in Them1 localization also occurred in brown adipose cells in vivo (Fig. 3). For this, mice were injected with saline (control) or with the β3 adrenergic receptor selective agonist CL316,243. In saline-injected mice, Them1 localized to distinct puncta near LD and were in the cytoplasm (Fig. 3a). The strong fluorescence signal for Them1 at baseline was due to mild cold exposure by housing mice at 22 °C, which is below their thermoneutral zone. Tissues at 4 h after saline injection showed the same localization of Them1 in puncta near lipid droplets and within the cytoplasm; Them1-containing puncta at 4 h after saline injection occupied a small intracellular volume (Fig. 3b, c).

By 1 h after CL316,243 administration, LD were reduced in size and number due to lipid hydrolysis and Them1 showed a larger cytoplasmic volume (Fig. 3c, d). By 2 h after CL administration, LD were less abundant and Them1 occupied a larger cytoplasmic volume compared to 1 h (Fig. 3c, e). By 4 h after CL316,243 administration, there were few intracellular LD, as evidenced by the loss and disruption of perilipin 1 expression, and Them1 occupied a larger cytoplasmic volume yet (Fig. 3c, f). To accompany the diffuse localization of Them1 from 1–4 h after CL administration was an increase in the nuclear localization of Them1 (Fig. 3d–f). When taken together with observations in vitro using iBAs, these in vivo studies support the notion that reorganization of Them1 puncta occurs after stimulation that activates the β3 adrenergic receptor (Fig. 2c).

**S-phosphorylation of the N-terminus of Them1 drives localization after stimulation**. To develop direct evidence that phosphorylation of N-terminal serine residues of Them1 are involved in puncta formation and dissolution, we deleted the first 36 amino acids of Them1 to generate a construct (Δ1-36-EGFP) that did not contain the phosphorylation sites that were detected in vivo or following transfection and stimulation of iBAs (Fig. 4a, b). The Δ1-36 mutant of Them1 exhibited a diffuse localization (Fig. 4c). We then linked a peptide containing the first 36 amino acids of Them1 directly to EGFP (Fig. 4b), which exhibited the same punctate localization as full-length Them1 (Fig. 4c). These results strongly support that amino acids 1–36 direct Them1 localization to puncta and suggest that phosphorylation of Them1 at one or more serine residues in the N-terminal region drives the conformational change in Them1 after stimulation.

To further explore this idea, we mutated each putative phosphorylation site by exchanging S for aspartic acid (S15-18-25D, or DDD), which mimicked the phosphorylated state of the S amino acid residues that are phosphorylated in vivo (Fig. 4b). This construct resulted in diffuse Them1 localization (Fig. 4d). When the S sites were mutated in pairs (Fig. 4e), or individually (Fig. 4f), S15D and S25D led to diffuse Them1 localization.

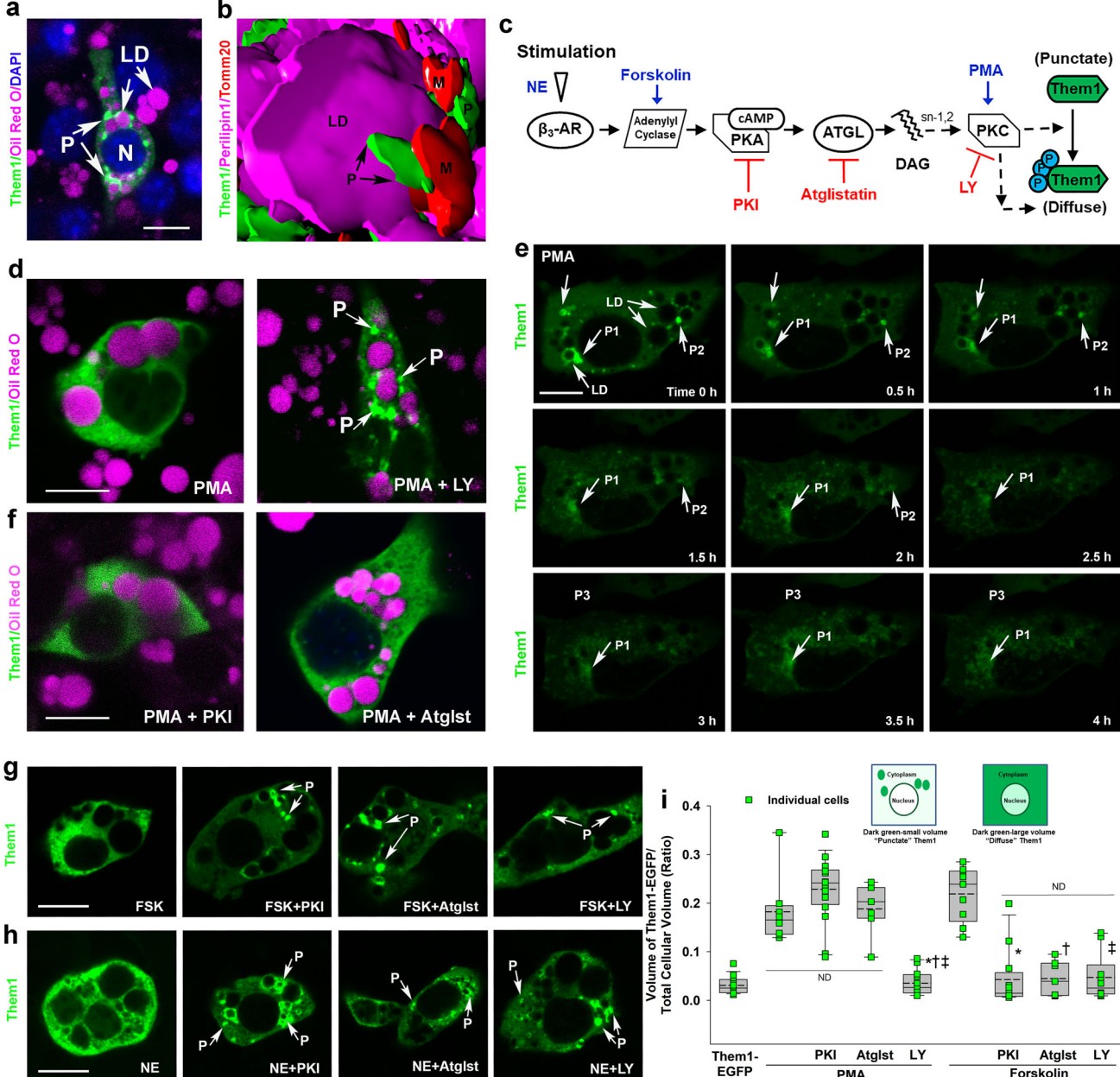

**Fig. 2 Regulation of Them1 subcellular localization in iBAs. a** Confocal microscopy of an iBAs expressing Them1-EGFP (green). LD lipid droplets (magenta), N nucleus (blue), P puncta (green). **b** 3-D image of an iBAs expressing Them1-EGFP (green), perilipin 1 to identify lipid droplets (LD, magenta), and Tomm20 to identify mitochondria (M, red). This image is shown in surface projection mode. **c** Schematic illustration of the putative Them1 regulatory pathway required for the phosphorylation and 3-D organization of Them1. β₃-AR beta-3 adrenergic receptor, DAG diacylglyerol, NE norepinephrine. DAG and PKC may have an indirect effect on puncta dissolution (dashed lines). **d** PMA-induced activation of iBAs expressing Them1-EGFP (green) resulted in the diffusion of Them1, which can be blocked by the PKCβ inhibitor LY333531 (LY). **e** Live-cell imaging was used to record changes in Them1 localization over time after PMA treatment. LD lipid droplets, P1–P3 individual puncta (green). **f** After PMA-induced PKC activation, neither PKI or atglistatin (Atglst) blocked the diffusion of Them1 from puncta. **g**, **h** Whereas the activation of PKA and PKC by forskolin (FSK) or norepinephrine (NE) resulted in the diffusion of Them1, PKI, Atglst, and LY blocked the diffusion of puncta (P, green). Scale bars in **a–h**, 10 μm. **i** Box and whisker plot with individual cell data (green squares). For each box, a solid line represents the median and a dashed line is the mean. Outliers are shown at the 5th/95th percentiles from 3 to 5 different experiments with the following n: Them1-EGFP, n = 16; PMA, n = 8; PMA + PKI, n = 16; PMA + Atglst, n = 7; PMA + LY333531, n = 12; FSK, n = 9; FSK + PKI, n = 12; FSK + Atglst, n = 7; FSK + LY333531, n = 10. For PMA, *P = 0.024 PMA + LY vs PMA alone; †P < 0.001 PMA + LY vs PMA + PKI; ‡P < 0.01 PMA + LY vs PMA + Atglst. For forskolin, *P < 0.001 forskolin + PKI vs forskolin alone; †P = 0.011 forskolin + Atglst vs forskolin alone; ‡P = 0.006 forskolin + LY vs forskolin alone. ND no difference. Statistical significance as determined by a multiple comparisons ad hoc test (Dunns Method) after Kruskal–Wallis one-way ANOVA on ranks.

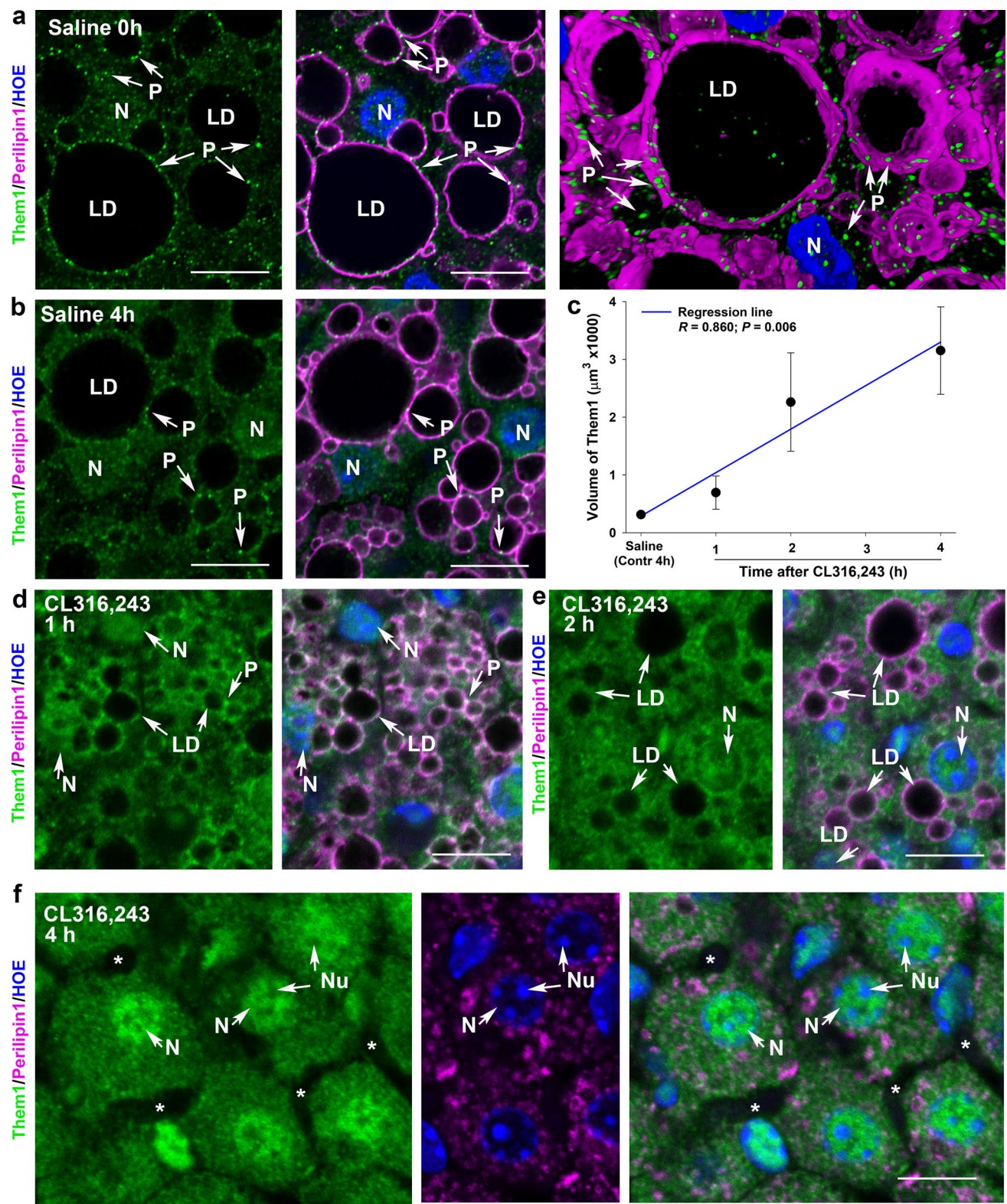

This result was quantified as the volume of Them1 per total cell volume using the strategy in Fig. 2i (Fig. 4g). By contrast, exchanging S for alanine at S15 or at S15, S18, and S25 (AAA, Fig. 4b), a configuration that cannot be phosphorylated, resulted in the punctate localization of Them1 alone or after stimulation with PMA, FSK, or norepinephrine (Fig. 4h). These results demonstrate that the phosphorylation of Them1 at S15 and/or S25 results in the diffuse localization of Them1 in iBAs.

**Puncta formed by Them1 are metabolically active structures.** Although plasmid-mediated transfection of Them1 constructs was sufficient for fluorescence-based imaging studies, the low transfection efficiencies prevented quantitative assessments of the impact of puncta formation on cellular metabolism. To obtain higher transfection efficiency, we constructed adenoviral vectors (Ad). These included wild-type Them1 or Them1 with AAA or DDD substitutions and EGFP fused at the carboxy terminus

**Fig. 3 Them1-containing puncta in brown adipose tissue (BAT) diffuse after stimulation in vivo. a** BAT was excised from mice immediately after the administration of saline. These tissues consisted of large lipid droplets (LD; magenta) surrounded by perilipin and numerous Them1-containing puncta (P; green) that were localized in the cytoplasm or in association with perilipin at the lateral edge of LDs. N, nuclei (blue) were stained with Hoechst 33342 (HOE). Scale bar, 10 μm. $n = 5$ images taken randomly from $n = 2$ mice with similar results. **b** BAT tissues from mice injected with saline and then tissues collected 4 h later. The localization of lipid droplets (LD, magenta), puncta (P, green), and nuclei (N, blue) were the same as in **a**. $n = 5$ images taken randomly from $n = 2$ mice with similar results. Scale bar, 10 μm. **c** Quantification of the 3-D volume of Them1 from mice injected with saline and collected 4 h later (contr; control) or CL316,243, a β3 adrenergic receptor agonist, at 1, 2, or 4 h after stimulation. Regression line indicates a positive and significant correlation between the volume of Them1 and time; the volume of Them1 is small when Them1 localizes to puncta in control cells and large as Them1 is diffuse after stimulation with CL316,243 (see Fig. 2i). Error bars represent mean ± SE from $n = 5$ images taken randomly from $n = 2$ mice/treatment and analyzed using Imaris software. Statistical significance was determined by ANOVA on the regression line, where $P = 0.006$. **d** BAT tissues from mice exposed to CL316,243 for 1 h. In these tissues, LD (magenta) were smaller than in saline-treated mice and Them1 (green) was more diffuse. N nuclei (blue). **e** By 2 h after CL316,243 administration, LD were less abundant and Them1 occupied a larger cytoplasmic volume. **f** BAT tissue from mice 4 h after the administration of CL316,243. LD (magenta) were not present, perilipin was cytoplasmic, and Them1 (green) was nearly completely diffuse in the cytoplasm. Over time (compare **d**–**f**) nuclear (N) Them1 staining (blue) increased in intensity. **d**–**f** $n = 5$ images taken randomly from $n = 2$ mice/treatment with similar results. Scale bars, 10 μm.

(Fig. 4a, b) with Ad-EGFP used as a control. Them1 expression at an MOI of 1:40 was ~45%, which resulted in Them1 expression similar to tissue expression in BAT of cold-exposed mice (Supplementary Fig. 2b). Both wild type (not shown) and AAA (Fig. 5a) substituted Ad-Them1-EGFP formed puncta in iBAs. Using near super-resolution imaging via Zeiss Airyscan, it was clear from 3-D images that Them1 resides in discrete puncta that were closely associated with LD (Fig. 5a). In contrast, DDD substituted Ad-Them1-EGFP exhibited a diffuse localization (Fig. 5b). In the near super-resolution 3-D images, diffuse Them1 filled the entire volume of cytoplasm not occupied by cellular organelles (Fig. 5b). These results overall were consistent with the results from our plasmid constructs.

We next examined the role of Them1 in regulating the oxygen consumption rate (OCR) in differentiated iBAs in response to norepinephrine stimulation. OCR values are a surrogate measure of oxidation rates of fatty acids generated by hydrolysis of LD-derived triglycerides[11]. After optimizing conditions of cell density by calculating the total number of EGFP-containing cells/total cells per plate (Supplementary Fig. 9a), OCR values were measured at baseline and after norepinephrine stimulation. Baseline and norepinephrine-stimulated OCR values were similar for Ad-DDD-EGFP and the Ad-EGFP control, whereas norepinephrine-stimulated values of OCR for Ad-Them1-EGFP and Ad-AAA-EGFP were reduced (Fig. 5c). These findings suggest that Them1 is active in suppressing LD-derived FA oxidation when in its punctate configuration. Because the stimulated redistribution of Them1-EGFP-APEX2 in puncta fully diffuses over a period of 4 h (Fig. 2e), increased OCR values within the first 1 h after norepinephrine exposure (Fig. 5c) would be expected to reflect only partial dissolution of Them1-containing puncta. By pre-incubating iBAs expressing Ad-Them1-EGFP with PMA prior to norepinephrine exposure, which would result in the further dissolution of puncta, we observed an increase in OCR values relative to no PMA (Fig. 5d). The addition of PMA to iBAs led to a progressive increase in values of extracellular acidification rate (ECAR) until the time of NE exposure in the absence of differences in OCR (Supplementary Fig. 9b). This result suggests that PMA stimulated anaerobic glycolysis, as opposed to fatty acid oxidation, which increases OCR following NE exposure. The PMA-induced increases in anaerobic glycolysis led to observed reductions in fatty acid oxidation, as reflected by OCR in response to norepinephrine stimulation, expressed as a percentage of baseline (Fig. 5e). When the effects of PMA treatment on OCR per se are taken into account, the NE-stimulated suppression of OCR on cells transduced with Ad-AAA was maintained compared to cells transduced with Ad-EGFP or Ad-DDD, (Fig. 5f and

Supplementary Fig. 9c). These results argue against downstream regulation of these protein constructs by PMA.

These metabolic studies support the role of Them1 puncta in regulating oxidative metabolism in BAT, which were further supported by measuring fatty acid oxidation contained within endogenous triglycerides in iBAs by pulse labeling with $^3$H-oleate and then determining triglyceride utilization during a chase period (Supplementary Fig. 9d). During the pulse period, equal fatty acid uptake was observed in iBAs transduced with Ad-EGFP, Ad-AAA-EGFP, or Ad-DDD-EGFP, leading to equal cellular accumulation of triglycerides (Supplementary Fig. 9d). By contrast, during the chase period triglyceride accumulated in iBAs transduced with Ad-AAA-EGFP, but not Ad-EGFP or Ad-DDD-EGFP (Supplementary Fig. 9d), which reflected reduced rates of fatty acid oxidation in the presence of Ad-AAA-EGFP (Supplementary Fig. 9e).

**Puncta are biomolecular condensates, or "membraneless organelles", by correlative light/electron microscopy.** To examine the ultrastructural details of puncta in iBAs, we next used Them1-EGFP vector constructs that included an ascorbate peroxidase–derived APEX2 tag appended to the carboxy terminus to perform correlative light/electron microscopy (Fig. 6). The APEX2 tag was developed with diaminobenzidine, resulting in brown reaction product only in cells expressing Them1 (Fig. 6a). This procedure allowed us to clearly determine which cells were transfected and expressing Them1. Them1 puncta were imaged using the AAA vector construct (AAA-EGFP-APEX2; Fig. 6b, c), and diffuse Them1 was evaluated using the DDD vector construct (DDD-EGFP-APEX2; Fig. 6d). Cells transfected with EGFP-APEX2 without Them1 showed pale staining uniformly throughout cells including the nucleus (not shown).

Consistent with fluorescence images of iBAs transduced with Them1-EGFP (Fig. 2a), puncta in electron micrographs of iBAs transduced with AAA-EGFP-APEX2 were found exclusively in the cytoplasm in close proximity to both LD and mitochondria (Fig. 6a–c). Although the puncta resembled mitochondria from steroid-secreting cells[15], there was no apparent membrane surrounding the structure. Instead, puncta were an aggregate of small round and elongate droplets with amorphous boundaries (Fig. 6b, c). Diffusely localized Them1 in iBAs transduced with DDD-EGFP-APEX2 was not clearly associated with any cytoplasmic structure (Fig. 6d).

Because puncta were membraneless structures with characteristics similar to that described for biomolecular condensates within the cell cytoplasm[16], we evaluated the Them1 primary sequence for characteristics that would facilitate a phase separation, aggregation, and/or the formation of puncta (Fig. 7).

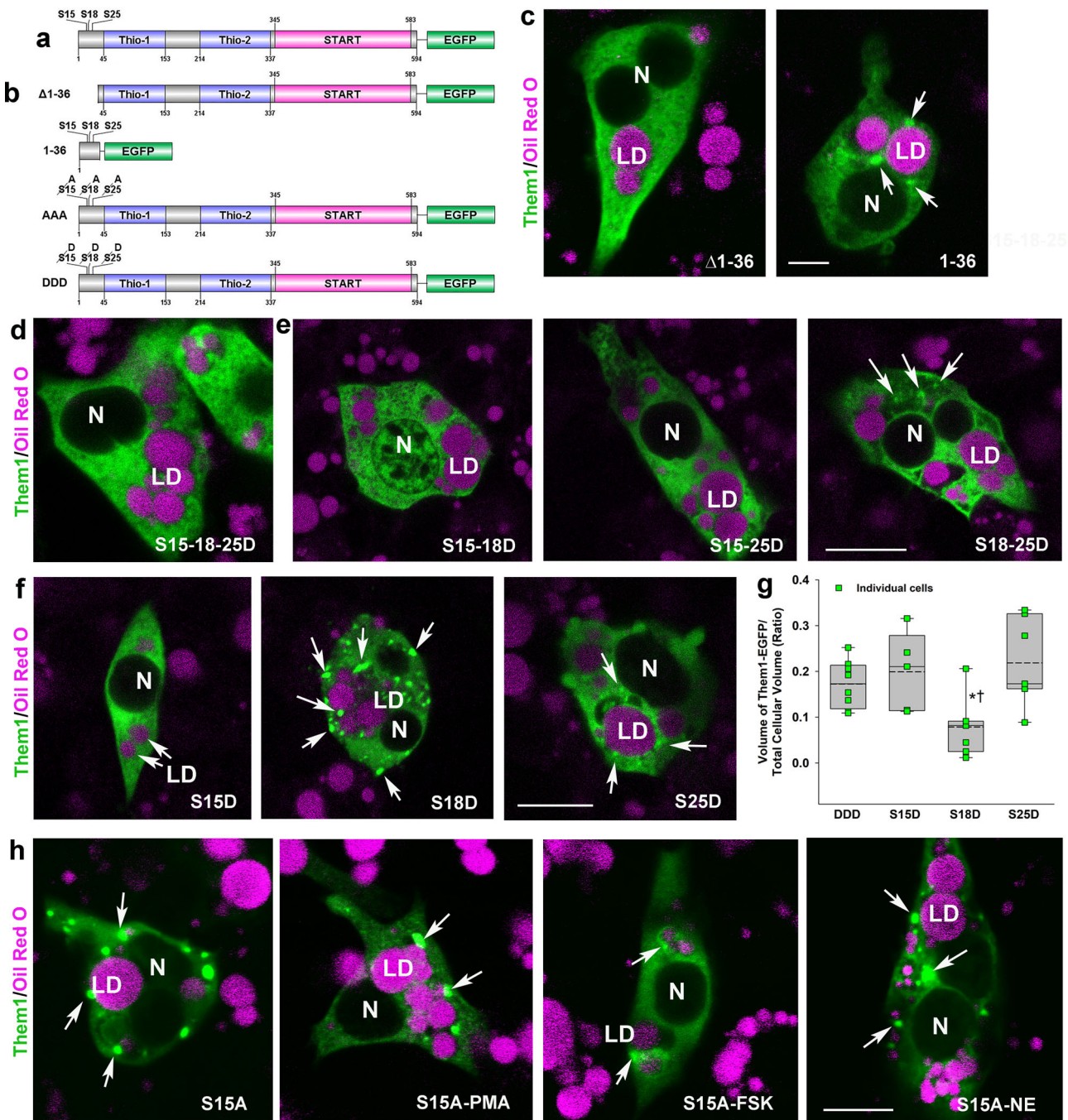

**Fig. 4 Phosphorylation at S15 and S25 in the N-terminus regulates the diffusion of Them1 in iBAs. a, b** Illustration of the Them1 structure and its mutant constructs tagged with EGFP used in the experiments. **a** Them1, highlighting the N-terminus S phosphorylation sites, linked to EGFP at the carboxy terminus. **b** Δ1-36, deletion of amino acids 1-36 in the Them1 sequence; 1-36, a construct containing only the first 36 amino acids in the Them1 sequence; AAA, serine residues at amino acids 15, 18, and 25 were mutated to alanine (A); DDD, serine residues at amino acids 15, 18, and 25 were mutated to aspartic acid (D). All mutant Them1 constructs were linked to EGFP at the carboxy terminus. **c** the Δ1-36 Them1 construct diffuses in the cytoplasm (green) whereas the 1-36 construct forms puncta (green) around lipid droplets (LD), which were stained with Oil Red O stain (magenta). **d** Mutation of S15/S18/S25 to aspartic acid (DDD) results in diffuse Them1 localization (green). **e** S15/S18 or S15/S25 mutant constructs result in diffuse Them1 localization (green) whereas the S18/S25 construct expressed some puncta and some diffuse Them1 (green). **f** The single mutation at S15 or S25, but not S18, to aspartic acid resulted in the diffusion of Them1 (green). **g** Box and whisker plot with individual cell data (green squares) to quantify volume measurements for mutation of S15/S18/S25 (DDD), or single mutations at S15D, S18D, or S25D. For each box, a solid line represents the median and a dashed line is the mean. Outliers are shown at the 5th/95th percentiles. $n = 2-5$ cells each from two different preparations. Data were evaluated using a one-way ANOVA with all pairwise multiple comparison using the Holm–Sidak method. *$P = 0.05$, S18D vs S15D; †$P = 0.011$, S25D vs S18D. **h** The serine to alanine mutation at S15 results in Them1-containing puncta, which are not dispersed after activation by PMA, FSK, or NE. Scale bar in **c–h**, 10 μm. For each construct evaluated in Fig. 4, the data are representative of $n = 3$ independent experiments with 3–5 individual cells photographed per treatment with similar results.

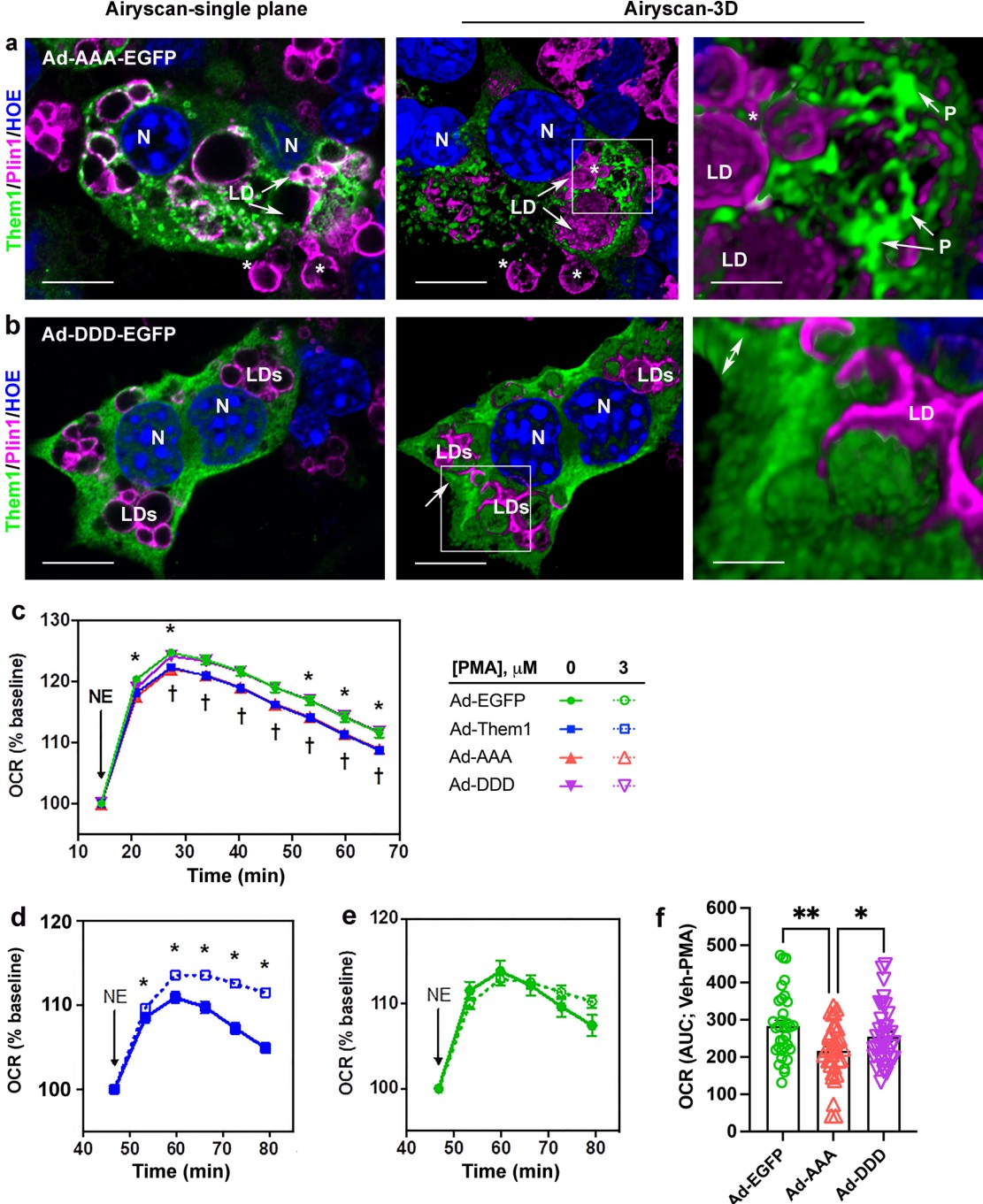

Bioinformatic analysis of the amino acid sequence for Them1 (Fig. 7a and Supplementary Fig. 10a) showed a highly disordered region at the N-terminus spanning residues 20-45 as predicted by IUPRED (prediction of intrinsic disorder) curves (Fig. 7b). There were no prion-like regions in the sequence as determined by PLD (Fig. 7c) or FOLD curves (Supplementary Fig. 10b). The N-terminus contains a high proportion of charged residues with a patch of basic residues followed by a patch of acidic residues (Supplementary Fig. 10c, d), which could engage in non-covalent crosslinking with other proteins or itself to drive a phase transition[17]. However, the PScore, a pi-pi interaction predictor, did not reach significance (Fig. 7d), which suggests that the sequence of Them1 does not have the propensity to phase separate based on pi–pi interactions. ANCHOR2 analysis, which predicts protein binding regions in disordered proteins, showed

that the N-terminal 20 amino acids were a highly disordered binding region (Fig. 7e). Interestingly, two of the phosphorylation sites, S15 and S18, lie in this region (Fig. 7a), suggesting that the phosphorylation sites may regulate binding events. Because multivalent interactions between proteins can lead to a phase separation[16], we also performed a motif scan using the Motif Scan server (https://myhits.isb-sib.ch/cgi-bin/motif_scan). This analysis highlighted no strong association of the 1st 20 amino acids with other proteins.

## Discussion

Them1 was first identified as a gene that is markedly upregulated in response to cold ambient temperatures and is suppressed by warm temperatures[9]. Whereas this initially suggested that Them1 function was to promote thermogenesis[9], homozygous disruption

**Fig. 5 The punctate localization of Them1 is metabolically active. a** Ad-Them1-EGFP with S15, S18, and S25 mutated to alanine (Ad-AAA-EGFP, green) formed puncta near lipid droplets (LD, magenta). This close relationship was highlighted by using near super-resolution Zeiss Airyscan imaging and reconstructing cells in 3-D. **b** Phosphomimetic mutations at S15, S18, and S25 (Ad-DDD-EGFP, green) caused diffusion of Them1. In 3-D, Them1 filled the thickness of the cytoplasm (double white arrow) and could be found above and below the plane of organelles including LD (magenta), which excluded Them1. N nucleus (blue). Scale bars, 10 μm (low magnification images-left and center); 2.5 μm (high magnification image-right). Data are representative of $n = 3$ independent experiments with similar results. **c** iBAs transduced with Ad-Them1-EGFP (blue), Ad-AAA-EGFP (red), Ad-DDD-EGFP (violet), or Ad-EGFP (green) were stimulated with norepinephrine (NE). The response of OCR values to stimulation was measured as % of baseline. Data ($n = 66–72$ wells from three independent experiments) were evaluated using mixed-effects analysis (group effect $P = 0.001$; interaction effect $P < 0.001$) followed by multiple comparison testing using the Tukey method (*$P = 0.015$, 0.034, 0.041, 0.039, 0.030, Ad-Them1-EGFP vs. Ad-EGFP; †$P = 0.031$, 0.040, 0.034, 0.018, 0.013, 0.009, 0.007, Ad-AAA-EGFP vs. Ad-DDD-EGFP). **d** OCR values for Ad-Them1 (blue) with or without PMA. Data were evaluated using mixed-effects analysis (group effect $P < 0.001$; interaction effect $P < 0.001$) followed by multiple comparison testing using the Sidak method (*$P = 0.012$, <0.001, <0.001, <0.001, 0 vs. 3 μM PMA). **e** OCR values for Ad-EGFP (green), Ad-AAA-EGFP (red), and Ad-DDD-EGFP (violet) with or without PMA. Data were evaluated using mixed-effects analysis (vehicle: group effect $P = 0.050$, interaction effect $P < 0.001$; PMA: group effect $P = 0.060$, interaction effect $P = 0.015$) with multiple comparison testing using the Holm-Sidak method (vehicle: #$P = 0.005$, 0.003, Ad-EGFP vs. Ad-AAA-EGFP, *$P = 0.034$, Ad-DDD-EGFP vs. Ad-AAA-EGFP; PMA: #$P = 0.032$, 0.024, 0.024, Ad-EGFP vs. Ad-AAA-EGFP). **f** Change in area under the curve (Δ AUC) for vehicle – PMA. Data ($n = 33–36$ wells from three independent experiments) were evaluated using a one-way ANOVA ($P = 0.004$) with multiple comparison testing using the Holm–Sidak method (*$P = 0.046$, Ad-DDD-EGFP vs Ad-AAA-EGFP, ##$P = 0.002$, Ad-EGFP vs Ad-AAA-EGFP. Data are mean ± SE.

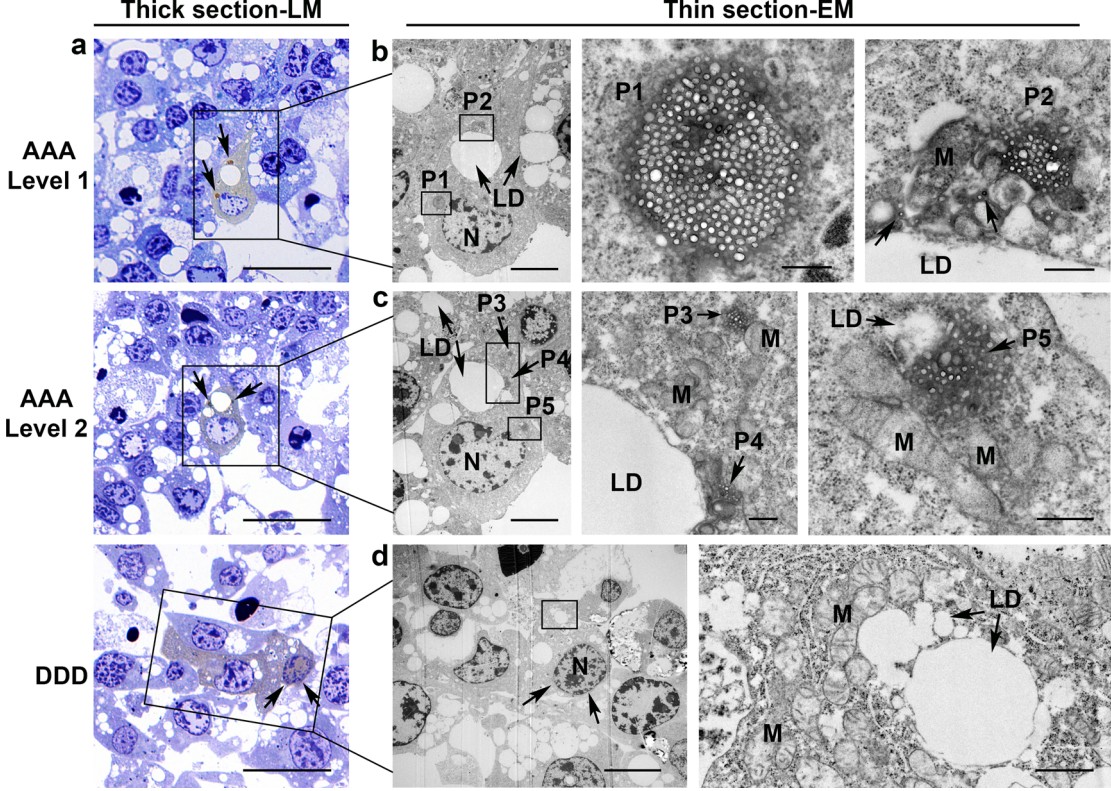

**Fig. 6 Correlative light and electron microscopy identifies puncta as biomolecular condensates. a** Thick, 0.5 μm, plastic sections of iBAs in culture transfected with phosphorylation mutants AAA- or DDD-Them1-EGFP-APEX2. Cells stained brown (box) also have dark brown punctate structures in mutant AAA-Them1 expressing cells (arrows). For AAA-transfected cells, serial sections show two different levels of the same cell. No puncta are present in cells transfected with the DDD-Them1 mutant although Them1 is found in the nucleus (arrows). Scale bar, 25 μm. **b** In thin sections of the boxed cell in **a**, puncta 1 (P1) is localized near the nucleus (N) and puncta 2 (P2) is localized near a lipid droplet (LD). At high magnification, punctate structures, P1 and P2, have no surrounding membrane and consist of liquid droplets embedded in an amorphous matrix. Scale bars, 5 μm (low magnification; original magnification, ×4000) and 0.5 μm (high magnification; original magnification, ×60,000). **c** In thin sections of the boxed cell in **a**, this section shows smaller puncta in the cytoplasm (P3), puncta associated with lipid droplets (LD; P4), and puncta in close association with LD and mitochondria (M) in the cell cytoplasm (P5). Scale bars, 5 μm (low magnification; original magnification, ×4000) and 0.5 μm (high magnification; original magnification, ×60,000). **d** In thin sections from the boxed region in **a**, no puncta are found in cells transfected with the DDD-Them1 mutant. Them1 is distributed throughout the cytoplasm and in the nucleus (N; arrows). Scale bars, 10 μm (low magnification; original magnification ×5000) and 1 μm (higher magnification; original magnification, ×25,000). All images are representative of the results obtained from $n = 3$ independent experiments.

of *Them1* in mice led to increased energy expenditure[10,11]. This result indicated that Them1 functions, on balance, to suppress energy expenditure. The current study helps to explain how a suppressor of energy expenditure can be induced by cold ambient temperatures, which also upregulates thermogenic genes; Them1 is phosphorylated and inactivated by the same adrenergic stimulus that promotes thermogenesis. Additionally, our studies have demonstrated phosphorylation-dependent intracellular

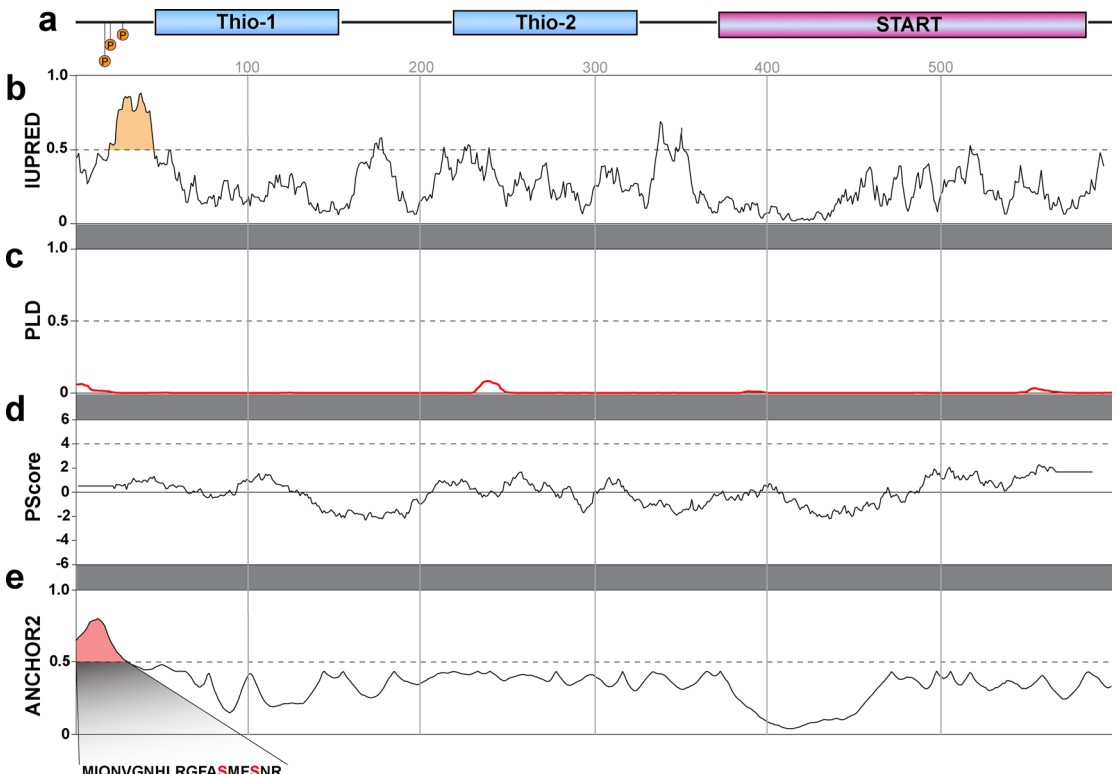

**Fig. 7 Bioinformatic analysis of Them1 suggests that a disordered region at the N-terminus spanning residues 20–45 may be involved in puncta formation.**
**a** Schematic diagram of the Them1 amino acid sequence aligned with the amino acid number from the N- to C-terminus. Thio indicates the position of thioesterase domains 1 and 2 and START indicates the position of the START domain. **b** Results from the IUPRED server, which predicts regions of disorder. The dashed line signifies the threshold for significance. The data indicate a large region of disorder in the first 50 amino acids (highlighted in gold). **c** Results from the Prion-like Amino Acid Composition server. The dashed line at 0.5 is a threshold for significance. The graph, red line, demonstrates no prion-like regions in the Them1 amino acid sequence. **d** Results from the PScore, which is a predictor of the propensity to phase separate based on pi–pi interactions. The dashed line is a threshold set at 4 to determine significance; the amino acid sequence of Them1 did not reach significance. **e** Results from an ANCHOR2 analysis, which predicts protein binding regions in disordered proteins. This analysis demonstrates that the N-terminal 20 amino acids of Them1 were strongly predicted to be a disordered binding region (highlighted in red). Two of the phosphorylation sites (red font) lie in this region (expanded region, also refer to Fig. 4b), suggesting they could potentially regulate a binding event.

redistribution of Them1 in response to thermogenic stimuli, suggesting a model for the regulation of energy expenditure within BAT. For this, Them1 activity is transiently silenced during peak thermogenesis by relocation away from its active site in puncta, which are located in close proximity to LD and mitochondria.

Our prior observations indicate that the activity of Them1 in brown adipocytes is to suppress the mitochondrial ß-oxidation of LD-derived fatty acids[11]. In keeping with this possibility, we observed that Them1 localized to puncta in the absence of adrenergic stimulation both in vitro and in vivo. These structures are juxtaposed with mitochondria and LD such that they could be expected to regulate fatty acid trafficking between these two organelles. Moreover, experiments in iBAs transduced with recombinant adenovirus revealed that full-length Them1 in unstimulated cells and the AAA-Them1 mutant, which is unresponsive to phosphorylation, were localized to puncta and were active in suppressing norepinephrine-stimulated oxygen consumption. These results are similar to our previous findings in cultured primary brown adipocytes, where Them1 actively suppressed norepinephrine-stimulated oxygen consumption[11], although puncta formation and dissolution were not investigated in that study.

By contrast, our data demonstrated that Them1 in its diffuse configuration was ineffective at suppressing oxygen consumption. This result was consistent with its relocation away from LD and

mitochondria and suggests that phosphorylation within the N-terminus amino acids 1–36, likely S15 phosphorylation, and to a lesser degree S25 phosphorylation, relocates and inactivates Them1 in vivo. Levels of S15 phosphorylation that were below detectable levels in our study could be due to the nature of peptide cleavage sites for mass spec analysis, glycosylation of nearby sites, or indicate that PMA-induced stimulation is not relevant to S15 phosphorylation, which may instead reflect norepinephrine stimulation. These interesting possibilities must be pursued in future studies. Phosphopeptide analysis identified other potential sites, including S27 and S31, the roles of which in Them1 localization remain unknown but also need to be studied experimentally. Our inhibitor data suggest that PKCβ plays some role in regulating Them1 metabolic function by changing the physical state of Them1. This may be by phosphorylating one or more Them-1-interacting proteins after stimulation or other yet undefined mechanisms. Interestingly, PKCβ appears to be involved in coordinating thermogenesis, as evidenced by increased rates of fatty acid oxidation in both BAT and white adipose tissue in mice globally lacking this kinase[18].

The important role played by Them1-containing puncta in inhibiting the mitochondrial oxidation of LD-derived fatty acids begs the question about the composition of puncta, including how they are formed and why they disappear after phosphorylation. Our data using correlative light and electron microscopy suggest that Them1 constitutively resides in biomolecular

condensates, also known as membraneless organelles, much like the nucleoleus, stress granules, and p-bodies[19,20]. Biomolecular condensates form by liquid-liquid phase separation or condensation[21], which favors aggregation. The resulting structure often resembles an aggregation of round liquid droplets[16], which is the result we found here for Them1 in puncta by electron microscopy. The driving force underlying the formation of biomolecular condensates is the exchange of macromolecular/macromolecular and water/water interactions under conditions that are energetically favorable[19]. What these conditions might be for Them1 are unknown but may be related to temperature, localized pH changes, or co-solutes that are expressed. Typical phase diagrams can define the conditions needed for condensation, which would be required to fully understand the localization of Them1 to puncta in vitro and in vivo.

One consistent signature of biomolecular condensates is the concept that their formation includes scaffold and client proteins[19]. Scaffold molecules are drivers of the condensation process and clients are molecules that partition into the condensate. It is not clear into which category Them1 fits, although the characteristics of its molecular structure suggest that it could be the primary scaffold protein. Phase separation of scaffold proteins and the subsequent partitioning of their clients are driven by network interactions for which two distinct classes of protein architecture are thought to be involved[19]. The first is multiple folded (SH3) domains that interact with short linear motifs[19]. Since Them1 does not have an SH3 domain, as determined by using the Simple Modular Architecture Research Tool web server[22], this could not be the sequence element driving puncta formation. Second is intrinsically disordered regions[19], for which Them1 has a strong candidate region near the amino terminus; amino acids 20–45 represent a highly disordered area that includes the S25 phosphorylation site and the N-terminal 20 amino acids are strongly predicted to be a disordered binding region that also includes phosphorylation sites at S15 and S18. Because the N-terminal 36 amino acids of Them1 are sufficient to drive puncta formation in iBAs, our data strongly support that this region is involved in puncta formation and should be further investigated. Alternatively, Them1 may function as a client. Clients often have molecular properties similar to scaffolds including intrinsic disorder[20], also making this role for Them1 a distinct possibility.

Dissolution of biomolecular condensates can be used to change the available concentration of cellular proteins to affect reaction efficiency[19], and is likely important for Them1 to reduce its local concentration at LD and mitochondria subsequently increasing fatty acid availability for β-oxidation in mitochondria and increasing thermogenesis. The phosphorylation state of proteins regulates their ability to assemble in biomolecular condensates, with threonine phosphorylation resulting in condensation and serine phosphorylation reducing retention in a dose-dependent manner[16]. Phosphorylation events were shown to regulate phase separation, aggregation, and the retention of mRNA's in FUS proteins, which are neuronal inclusions of aggregated RNA-binding protein fused in sarcoma[23,24]. It is thus phosphorylation at S15 and S25, which are highly conserved residues in Them1 that reduces its retention in puncta as evidenced in inhibitor studies and by genetic manipulation of Them 1. Norepinephrine-induced protein phosphorylation events during cold exposure in addition to a 3-fold increase in Them1 protein concentration over time[11], likely creates equilibrium between active Them1 in puncta and inactive Them1 that is diffusely localized in the cytoplasm. Our data thus suggest that the delicate balance of Them1 protein concentration, its phosphorylation status, and environmental factors that promote Them1 phase transitions dictate the rate of thermogenesis in BAT.

Together, our results reveal an unexpected mechanism for the regulation of Them1, a protein that reduces fatty acid availability for β-oxidation in mitochondria and limits thermogenesis. Manipulation of Them1 localization in BAT may provide a therapeutic target for the management of metabolic disorders including obesity and non-alcoholic fatty liver disease.

## Methods

**Culture of brown adipocytes.** Immortalized brown preadipocyte cells (iBAs) from mouse brown adipose tissue[25] were obtained from Dr. Bruce Spiegelman. iBAs were cultured in Dulbecco's Modified Eagle's medium (DMEM; Gibco, Grand Island, NY), 20% fetal bovine serum (FBS; Gibco), 1% penicillin/streptomycin solution (P/S; Gibco), and 2 μM HEPES (Gibco). Confluent iBAs were induced to differentiate from preadipocytes by adding induction cocktail containing 5 μM dexamethasone (Sigma-Aldrich, St. Louis, MO), 125 μM indomethacin (Sigma-Aldrich), 0.02 μM insulin (Sigma-Aldrich), 500 μM isobutylmethylxanthine (Life Technologies, Carlsbad, CA), 1 nM triiodothyronine (Sigma-Aldrich), and 1 μM rosiglitazone (Sigma-Aldrich) in medium consisting of DMEM, 10% FBS, and 1% P/S. After 48 h of induction, the medium was replaced by maintenance medium consisting of DMEM, 10% FBS, 1% P/S with 0.02 μM insulin, 1 nM triiodothyronine, and 1 μM rosiglitazone. The maintenance medium was replaced every 2 days. For immunostaining and confocal applications, iBAs were plated on glass coverslips, which were pre-treated with 0.01% collagen type I (Sigma-Aldrich) in 0.01 M acetic acid.

**Production of DNA constructs.** The full-length cDNA for Them1 (from *Mus musculus*: Q8VHQ9) was cloned and linked to EGFP plus an engineered ascorbate peroxidase (APEX2) fusion protein the C-terminus. Codons for S15, S18, and S25 were mutated to alanine (A) or aspartic acid (D) using the QuikChange mutagenesis kit (Agilent, Santa Clara, CA). The Δ1-36 mutation includes deletion of the first 36 amino acids at the N-terminus of Them1, whereby Them 1 begins at amino acid 37 (M; AUG). The full-length cDNA for Them1 was cloned into a pVQAd CMV K-NpA vector (ViraQuest, North Liberty, IA) along with C-terminal EGFP downstream of the CMV promoter. The Them1 adenovirus was created by ViraQuest.

**Heterologous Them1 expression in cultured iBAs.** Plasmid transfection with Lipofectamine 3000 (Thermo Fisher Scientific, Waltham, MA) was conducted 48 h after iBAs were induced to differentiate. Purified DNA and P3000 reagent were mixed with Opti-MEM. The mixture was then added into Opti-MEM containing Lipofectamine 3000. The three reagents were allowed to incubate at room temperature for 10 min and then added to the cells. After 7 h of transfection, the medium was replaced by maintenance media. After 48 h, the cells were assayed. The same ratio of cells to DNA was used for each experiment but due to the very low transfection efficiency, it was not possible to assay Them1 concentration by immunoblotting.

For experiments using adenoviral vectors to express Them1, recombinant adenoviruses including Ad-Them1-EGFP, Ad-AAA-EGFP, Ad-DDD-EGFP, or Ad-EGFP were added to the medium for 24 h after day 1 of differentiation. Adenoviral vectors were initially screened for MOI versus cell protein expression using our Them1 antibody[11], mRNA expression using quantitative RT-PCR (Supplementary Fig. 3b, c), and cell viability using the crystal violet assay[26], which was nearly 100% with MOI's ranging from 10 through 80 (not shown). Unless otherwise specified, an MOI of 40 was used because Them1 expression was most similar to that induced in BAT during cold exposure (Supplementary Fig. 3b).

**LC-MS/MS for detecting Them1 phosphorylation.** Protein concentrations were estimated with a Bradford protein assay prior to acetone precipitation, done as described[27]. Pellets were resuspended in 8 M urea (Proteomics grade, PlusOne; GE Healthcare, Chicago IL), 5 mM TCEP (Bond-breaker; Thermo Fisher Scientific, Waltham, MA), 50 mM triethyl ammonium bicarbonate (TEAB, Sigma-Aldrich) and then alkylated with 100 mM iodoacetamide to a final concentration of 9 mM. Samples were then diluted with 100 mM TEAB (Sigma-Aldrich) followed by incubation with trypsin (Sigma-Aldrich) at a ratio 1 : 40 trypsin:total protein overnight at 37 °C. Samples were acidified to a final concentration 0.1% TFA (Sigma-Aldrich) and then desalted using HLB solid phase extraction plates (Waters, Milford, MA). Eluted peptides were dried in a centrifugal vacuum concentrator and then stored at −20 °C.

Phosphorylated peptides were enriched with Fe-NTA magnetic beads (Cube Biotech, Cambridge, MA) using a 96-well plate magnet[28]. Beads were washed and the phosphorylated peptides eluted with 50% acetonitrile and 1% ammonium hydroxide. Eluted peptides were then acidified with formic acid and dried. Before LC-MS/MS analysis, samples were resuspended to yield a 0.3 μg/μL solution, assuming a 1 : 100 reduction in peptide amount.

LC-MS/MS analysis was conducted with a 1200-bar nano HPLC (Thermo Easy 1200 nLC; Thermo Fisher Scientific) using a 15 cm × 75 μm ID, 1.6 μm C18 column (Ionoptickcs, Aurora series emitter; Fitzroy VIC, Australia). The temperature of the column was maintained with an integrated column oven (PRSO-V1; Sonation lab

solutions, Biberach, Germany). Enriched phosphorylated peptides were chromatographed using a mixed linear gradient of buffer A containing 0.1% formic acid in water and buffer B containing 0.1% formic acid in 80% acetonitrile. Flow rates were maintained over the course of the HPLC run and the gradient applied was (Time:%B): 0:3, 1:7, 81:30, 91:40, 93:80, 96:80, 97:3. The LC system was interfaced to an Eclipse Tribrid Orbitrap mass spectrometer (Thermo Fisher Scientific). The positive spray voltage was set to 2–2.5 kV, ion transfer tube temp 305 °C, RF funnel 30%. The FAIMS Pro system was set to survey three compensation voltages of −40, −60, and − 80 V. The full profile MS spectra were collected at 120k resolution (200 m/z). The data-dependent acquisition used a monoisotopic peak determination filter set to peptides, intensity set to 5e3, charge states 2–6, dynamic exclusion of 60 s and a cycle time of 0.6 s for each FAIMS CV used, total cycle time 1.8 s. The MS/MS fragmentation was collected with high energy collision induced dissociation with the following settings; quadrupole isolation with 1.6 Da window, normalized collision energy set to 30%, detector set to ion trap with rapid scan rate and defined mass range of 200–1500, AGC 250% with a maximum injection time set to 35 ms data stored as centroid data.

A pooled sample of enriched peptides from all of the samples was crudely fractionated with centrifugal columns following manufacturer's instructions (Pierce High pH reversed-phase peptide fractionation kit (Thermo Fisher Scientific). This generated 8 fractions, a flow through, and wash fraction all were analyzed using the same method as the samples. The results were used to generate an in-depth library of phosphorylated peptides.

Data were processed using Proteome Discoverer v2.4 (Thermo Fisher Scientific). Data-dependent data were searched with Sequest module using the following parameters; MS1 mass accuracy 10 ppm with MS/MS tolerance 0.6 Da, searched against the *Mus musculus* protein database Uniprot taxonomy ID 10090 v2017-10-25, trypsin set as the digestion enzyme with maximum of three missed cleavages. Dynamic modification included oxidation of methionine, phosphorylation of Ser and Thr, N-terminal pyro-glutamate formation, N-terminal Met loss and Met loss with acetylation, static modification was set to cabamidomethyl of Cys. Validation was done using the percolator processing node with FDR set to 0.01. Them1 (Q8VHQ9) phosphorylated peptides identified via Sequest were imported to Skyline for chromatographic extraction and integration. The signal intensity of Them1 peptides was normalized to the phosphorylated peptide RSS(phospho)QGVLHMPLYTSPIVK from isoform 2 of hormone-sensitive lipase (P54310-2). This peptide was chosen as a housekeeping phosphorylated peptide. The total amount of Them1 phosphorylation was normalized to this peptide.

**Fixation and Oil Red O staining for LD**. Differentiated iBAs were washed with phosphate buffered saline (PBS) and then fixed with 4% paraformaldehyde in PBS for 20 min at room temperature. Cells were washed twice before staining with 0.3% Oil Red O solution in 60% isopropanol (Sigma Aldrich) at room temperature. The cells were washed with PBS and then mounted with ProLong Diamond Antifade Mountant with DAPI (ThermoFisher) to identify nuclei.

**Immunostaining for LD, mitochondria, and plasmid-expressed Them1**. Differentiated iBAs were fixed, as above, and then stained for perilipin (anti-Plin1) to identify LD, Tomm20 (anti-Tomm20) to detect mitochondria, or anti-EGFP to visualize the plasmid-transfected Them1-EGFP. For staining Them1 in paraffin sections of BAT, tissues were treated with Image-iT™ FX Signal Enhancer (Thermo Fisher Scientific, Hampton, NH) after antigen retrieval with citrate buffer, pH 6.0. The sections were stained with anti-Them1 antibody, anti-Plin1, and Hoechst 33342 to identify nuclei.

Details of the antibodies: affinity purified rabbit anti-Them1[11] 1:250 for immunostaining or 1:1000 for immunoblots (see Supplementary Fig. 5); anti-Tomm20, Santa Cruz Biotech (Dallas TX, USA), rabbit, sc-11415, 1:200; anti-Plin1, Fitzgerald (Acton MA, USA), guinea pig, 20R-PP004,1:500; anti-EGFP, Novus Biologicals (Littleton CO, USA), goat, NB100-1678, 1:500; anti-actin, Abcam (Cambridge, MA, USA), mouse, Ab3280,1:1000. Secondary antibodies labeled with HRP, AlexaFluor 488, 647, or Cy3 were purchased from Jackson ImmunoResearch (West Grove PA, USA).

**Confocal microscopy**. The localization of fluorescence signal in cultured iBAs was evaluated using a Zeiss LSM880 confocal microscope system with or without Airyscan. Images were acquired at high resolution as 2 µm z-stack slices through the thickness of cells and assembled in 3-D using Volocity image processing software.

For confocal imaging of living cells, a time-lapse image was created to capture the dynamics of protein expression and localization, signal transduction, and enzyme activity. Cells grown on coverslips were placed in the incubator chamber of a Zeiss LSM880 inverted confocal microscope system to visualize EGFP expression (linked to Them1). The interior of the chamber was kept at a constant 37 °C, and set to a 5% CO$_2$ level and a 95% O$_2$ level. An image was captured at T0, and immediately following the application of a treatment. The microscope then automatically tracked, focused, and photographed cells every 30 min for 4–5 h. These images are also taken in the high resolution z-stack format and analyzed using Volocity software.

**Treatment of iBAs to explore metabolic pathways**. To explore the metabolic pathway leading to reorganization of intracellular Them1, various treatments were applied for 4 h to cultured iBAs as follows: Norepinephrine (1 µM; Sigma Aldrich; St. Louis, MO), a neurotransmitter, was used to mimic cold exposure. Forskolin (1 µM; Selleckchem, Houston, TX), a membrane permeable labdane diterpene produced from the *Coleus* plant, was used to activate adenylyl cyclase. This treatment, through the activation of cAMP, activates PKA. PMA (3 µM; LC Laboratories, Woburn, MA) was used to activate PKC. NE, adenylyl cyclase, and PKC were selected for activation as they represent three key control points in the pathway allowing the characterization of all upstream regulators. Pathway inhibitors were added 1 h prior to activation and were continued throughout the 4 h activation period as follows: PKI [14–22] myristoylated (0.5 µM; Invitrogen, Camarillo, CA), a synthetic peptide inhibitor of PKA, was added to inhibit PKA activation. Atglistatin (40 µM; Selleckchem, Houston, TX) was used to selectively inhibit the processing of triacylglycerol to fatty acids via the formation of diacylglycerol by ATGL. Ruboxistaurin, or LY333531, (2 µM; Selleckchem, Houston, TX), an isozyme-selective inhibitor of PKC that competitively and reversibly inhibits PKC-βI and PKC-βII, was added to inhibit PKCβ activity.

**Correlative light and electron microscopy of Them1 in iBAs**. Differentiated iBAs transfected with a plasmid containing Them1 linked to APEX2 were fixed and the APEX2 developed using an ImPACT peroxidase substrate kit (Vector Labs, Burlingame, CA). Cells were then frozen using a Wohlwend Compact 02 High Pressure Freezer. Frozen cells underwent super-quick freeze substitution and embedding using the protocol established by McDonald[29]. In 0.5-µm thick sections, positive cells stained brown by light microscopy. When positive cells were identified, four consecutive levels of serial ultra-thin sections were obtained with one thick section for light microscopy between levels. Light microscopy images, taken with an Axioimager (Zeiss) equipped with a color ccd camera were used to follow puncta in the sections and to determine the position of positive cells in ultra-thin sections. Images were taken on a JEOL 1400 transmission electron microscope (TEM) equipped with a Gatan (Pleasanton, CA) Orius SC1000 camera. Low magnification TEM images were taken at an original magnification of ×4000–5000, intermediate magnification TEM images at ×25000, and high magnification TEM images at ×60,000.

**O₂ consumption rates in iBAs**. Oxygen consumption rates (OCR) were measured in iBAs using a Seahorse XFe96 (Agilent Technologies, Santa Clara, CA, USA). Briefly, iBAs were seeded at a density of 1000 cells/well in collagen-coated Seahorse XF96 cell culture microplates 2 days before induction. On day 1 after induction, iBAs were incubated with Ad-Them1-EGFP (MOI 40), Ad-AAA-EGFP (MOI 50), Ad-DDD-EGFP (MOI 60), or Ad-EGFP (MOI 40). In preliminary experiments these MOIs led to similar Them1 and EGFP protein expression. One day after adenovirus infection, the medium was changed to the maintenance medium supplemented with 1 µM NE. After 2 days, CO$_2$ was withdrawn from the cultures for 1 h at 37 °C in Krebs-Henseleit buffer (pH 7.4) containing 2.5 mM glucose, 111 mM NaCl, 4.7 mM KCl, 2 mM MgSO$_4$-7H$_2$O, 1.2 mM Na$_2$HPO$_4$, 5 mM HEPES, and 0.5 mM carnitine (denoted as KHB). Basal OCR was measured in KHB for 18 min and then 10 µM NE was injected through the Seahorse injection ports to a final concentration of 1 µM in each well. NE responses were measured after injections as previously described[11].

Relative transfection efficiency was assessed by counting the number of EGFP expressing cells relative to the number of total viable cells. Data were normalized to the number of viable cells, determined using the NucRed™ Live 647 ReadyProbes™ assay (Thermo Fisher Scientific, Waltham, MA, USA) and the color read using a Spectramax i3x (Molecular Devices, San Jose, CA, USA) visualized and calculated using the Spectramax i3x.

**Fatty acid oxidation**. Rates of oxidation of triglyceride-derived fatty acids were determined by pulse-chase methods as described by Cooper et al.[30]. In brief, iBAs were cultured, differentiated, and transfected in 12-well plates as described above. On day 2 prior to pulse, the media was changed to pre-labeling medium including DMEM, 10% FBS, 1 g/l glucose, and 10 mM HEPES. Cells were pulsed for 1.5 h with pre-labeling media containing 1 mM carnitine and 500 µM sodium oleate (Sigma-Aldrich) containing 2 µCi/ml [9,10-³H]oleate (Perkin Elmer, Waltham, MA) conjugated to BSA[31]. Cells were chased by washing with PBS containing 1% fatty acid-free BSA and then incubated for 1.5 h with pre-labeling media containing 1 mM carnitine, 500 µM oleate, and 5 µM NE. At the end of the chase period, cells were washed with PBS containing 1% fatty acid-free BSA prior to analysis. Relative transfection efficiency was determined by immunoblot analysis of Them1 and EGFP.

Lipids were extracted using the Folch method and then separated by thin-layer chromatography using hexane:ethyl-ether:acetic acid (80:20:1, v/v/v) to determine radiolabeled triglycerides[31]. The concentrations of total triglycerides were determined enzymatically (Wako Diagnostics, Mountain View, CA)[32]. Rates of fatty acid oxidation were determined from utilization of triglycerides during the chase period and expressed relative to Ad-EGFP.

**Activation of BAT in mice**. Male C57BL/6J mice (7 w) were purchased from the Jackson Laboratory (Bar Harbor, ME) and housed in a specific pathogen-free AAALAC International Facility with a standard 12 h alternate light/dark cycle at an ambient temperature of $22 \pm 2\,°C$ and 30–70% humidity, at Weill Cornell Medical College. Animal use and euthanasia protocols were approved by the Institutional Animal Care and Use Committee at Weill Cornell Medical College.

For the experiment, mice were moved to Promethion metabolic cages (Sable Systems International, Las Vegas, NV) with one mouse/cage. Mice were housed at 22 °C for the first 24 h, but were changed to 30 °C overnight prior to experiments. Room temperature, or 22 °C, is below the thermoneutral zone for mice and the mild cold exposure stimulates Them1 expression in BAT[11]. Experimental mice were injected subcutaneously with one dose of CL316,243 (Sigma-Aldrich, 1 mg/kg body weight) in saline, which is a β3 adrenergic receptor agonist. Control mice were injected with saline alone. Subscapular BAT was excised from mice after euthanasia at T0, 1, 2, or 4 h after injection. Tissues were fixed overnight in 10% neutral buffered formalin and then processed. Paraffin sections were used for H&E stained samples and for immunocytochemistry as described above.

**Image processing and quantification of fluorescence signal**. For experiments aimed to quantify changes in intercellular Them1 localization after stimulation, image segmentation was used to calculate the volume of Them1 by thresholding the green (EGFP) pixels of specified intensity that represented Them1-EGFP minus background intensity using either Velocity software (Quorum Technologies, Ontario, Canada) or Imaris software (Oxford Instruments, Concord, MA, USA). The strategy used for both iBAs and for tissue Them1 expression was that puncta represent concentrated Them1, which occupies a small intracellular volume whereas diffuse Them1 is distributed evenly throughout the entire cell volume resulting in a large intracellular volume (Fig. 1k, schematic diagram). Them1 fluorescence signal, which represented pixel intensity in the EGFP (green) channel, was measured by using a normalized histogram at each 12-bit (0–4096) intensity value. For iBAs, background signal was determined from cells that were not transfected and the fluorescence threshold set to exclude this low-intensity background signal. For paraffin sections of brown adipose tissue, the background fluorescence intensity was determined using Them1-deficient mice and green pixels in this range were excluded from the experimental data. This strategy did not stratify the signal into intense (as found in puncta) or weak (as found in the diffuse state), but instead identified the volume of total intracellular Them1 fluorescence irrespective of intensity.

**Bioinformatics**. The amino acid sequence of Them1 from *Mus musculus* was acquired from UniProt[33] and analyzed with multiple servers. The IUPRED2A server was used to predict intrinsic disorder and ANCHOR binding regions[34–37]. The Prion-like Amino Acid Composition (PLAAC) server was used to predict prion-like regions through the prion-like domains (PLD) and FOLD curves[38]. The propensity of Them1 to phase separate through long-range planar pi-pi contacts was predicted using a server generated by the laboratory of Professor Julie Forman-Kay (PScore)[39]. The Motif Scan server (https://myhits.isb-sib.ch/cgi-bin/motif_scan) was utilized to examine the first 20 amino acids for known motifs[40].

**Statistics and reproducibility**. Data were analyzed using SigmaPlot software with a $P$ value $< 0.05$ considered significant. Prior to analysis, data was subjected to an outlier test (Grubbs test; GraphPad on-line calculator https://www.graphpad.com/quickcalcs/grubbs1/) and individual data points were excluded that were extreme outliers, alpha value of 0.01 or less. For comparison of data from many groups, one-way analysis of variance was used to determine statistical significance. If variances were not normal, non-parametric ANOVA was performed using ranks. For experiments with multiple times and treatments, 2-way analysis of variance was used to determine statistical significance. Post-hoc analyses were done to determine differences between groups.

**Reporting summary**. Further information on research design is available in the Nature Research Reporting Summary linked to this article.

## Data availability

The authors declare that the main data supporting the findings of this study are available within the article and its Supplementary Information files. Adenoviral vectors or plasmid constructs are available upon reasonable request to the corresponding authors. The raw data files for LSMS were deposited at the Center for Computational Mass Spectrometry, Center for Computer Science and Engineering, University of California, San Diego. The dataset files can be accessed under the dataset ID MSV00008675 [https://massive.ucsd.edu/ProteoSAFe/dataset.jsp?task=cc447e32ff414e9f8002deaa8c5566eb]. Source data are provided with this paper.

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

## Acknowledgements

The authors thank Dr. Bruce Spiegelman, Dana-Farber Cancer Institute, Harvard Medical School for providing the iBAs, Mr. Kyle Smith for his technical help with electron microscopy, and Mr. Aniket Gad for expert advice on image processing. This work was supported by the National Institutes of Health (RO1 DK 103046 to D.E.C., S.J.H., and E.O.; R01 DK048873 to D.E.C.), the Harvard Digestive Diseases Center (P30 DK034854 to S.J.H.), and the National Institutes of Health shared-instrumentation grant program for the High Pressure Freezer (S10 OD019988-01 to S.J.H.). H.T.N is the recipient of a Pinnacle Research Award from the AASLD Foundation and T.I.K. is the recipient of a Weill Cornell Department of Medicine Pre-Career Award and acknowledges support from NIH T32DK007533. N.I. is the recipient of an America Heart Association Postdoctoral Fellowship and S.G. was supported by a Research Science Institute/Center for Excellence in Education Summer Research Fellowship.

## Author contributions

Y.L. did the cell culture experiments and all confocal microscopy experiments; N.I. did OCR studies without PMA; H.T.N. and T.I.K. performed the animal experiments, H.T.N. also conducted OCR studies with PMA and lipid oxidation experiments; B.R.R. and A.M.R. conducted the LC-MS/MS studies; M.B. assisted with experiments; L.H.A. did the immunostaining; M.T. and E.O. did the Them1 bioinformatic analysis; D.E.C. co-mentored Y.L., assisted with experimental design, and edited the paper. S.J.H. did CLEM, co-mentored Y.L., assisted with experimental design, and wrote the paper. All authors reviewed and edited the final manuscript.

## Competing interests

The authors declare no competing interests.
