## [Peer Review File · Nature Communications]

REVIEWER COMMENTS

Reviewer #1 (Remarks to the Author):

Review Li et al.

The manuscript by Li et al. describes the importance of Them1 phosphorylation in the regulation of thermogenesis. Them1, a BAT-enriched long chain fatty acyl-CoA thioesterase, is upregulated by cold exposure and reduces fatty acid availability for β -oxidation in mitochondria to limit thermogenesis. However, the detailed molecular mechanisms of Them1 regulation remain unclear. The author showed that Them1-GFP localized in puncta nearby LD and mitochondria and that β -adrenergic stimulation diffuses Them1 into cytoplasm, likely because phosphorylation of Them1 by PKC β .

Although potentially of interest, the physiological significance of Them1 phosphorylation with respect to thermogenic regulation in vivo is largely unclear. Furthermore, definitive evidence of a change in phosphorylation of Them1 during thermogenic signaling is lacking. The comments below address some of the specific weaknesses of the present study.

Major points:

1. The authors conclude that adrenergic stimulation induces serine phosphorylation of Them1 based on only Ala or Asp substitution and public database. This claim is vital to the manuscript. However, no definitive evidence of change in Them1 phosphorylation by adrenergic stimulation is provided. The authors should perform phosphorylation assay (i.e. ^{32}P i radiolabeling followed by pull down assay) to prove the dynamics of Them1 phosphorylation during adrenergic stimulation in iBA by comparison of Them1(WT) and Them1(AAA), at least.
2. Throughout the manuscript, the authors use EGFP-fusion Them1. They concluded that aggregation of Them1 was not due to EGFP, as evidenced by the diffused distribution of EGFP itself (Supplementary Figure 3a). However, EGFP forms a dimer at high concentration. They cannot rule out the possibility that localization of Them1 in certain region/organelle led to the formation of artificial aggregates of Them1-EGFP. In order to avoid artificial effect of tag on Them1 distribution, the authors should perform the same experiment in Figure 1a, b, d, f and Figure 3d, g on exogenously expressed untagged Them1 stained by anti-Them1 antibody they used in Figure 3 and Supplementary Figure 3b. The author should also perform EM study in Figure 5 using Them1-APEX2 without EGFP.
3. The sentence “~ Them1 was phosphorylated by PKC β at S15, ~” in abstract (page 2, line 10-11) and the scheme in Figure 3h are too assertive. The authors used PKC β inhibitor LY333531 to substantiate the putative involvement of PKC β . However, they failed to show Them1 distribution under PKC β knockout or knockdown condition. Furthermore, they didn't perform any in vitro kinase assay to show PKC β -dependent direct phosphorylation of Them1. If these experiments are too technically difficult, the authors should refrain from strong conclusions about the involvement of PKC β in Them1 translocation.
4. The claims of PKC β involvement contradict the literature. According to the authors' hypothesis, PKC β -depletion promotes Them1-punctate formation and suppress β -oxidation. However, PKC β ^{-/-} mice exhibit increased energy expenditure as a result of increased mitochondrial fatty acid oxidation in brown adipose tissue (Bansode RP et al., JBC. 2007, PMID: 17962198 and Huang W et al., JLR. 2012, PMID: 22210924). How do the authors explain this discrepancy with their results? They should discuss this issue.
5. The data shown in Figure 2 are crucial to show the translocation of Them1 in vivo. However, the possibility of non-specific staining cannot be excluded. Thus, the author should show the specificity of anti-Them1 antibody using BAT from Them1 KO mice. Furthermore, the results shown in Figure 2

show the mixed effect of both increased body temperature and CL injection. The authors acclimate mice to thermoneutral or optimized temperature to stabilize Them1 expression before CL316243 injection.

6. The title of this manuscript is too vague. It should be added that this study focused on brown adipocytes.

Minor points:

1. In Figure 1a, b, the authors should also stain other organelles such as lysosome, peroxisome, Golgi apparatus and ER.
2. Figure 3c-g needs statistics. The authors need to measure Them1-punctate volume in multiple cells, to discuss the contribution of each serine residue to Them1 distribution.
3. Page 6, line 21; “~concentration (Supplemental Fig 3)” should be “~ concentration (Supplemental Fig 4)”
4. Page 10, line 5 and line 9; The author need to be clear which Them1 mutant was used (i.e. ~ puncta in Them1(AAA)-EGFP-APEX2 cells were ~).

Reviewer #2 (Remarks to the Author):

Comments of the reviewers:

In this manuscript, Yue Li et al. investigate the role the Thioesterase Superfamily Member 1 (Them1) plays in thermogenesis in brown adipose tissue. The authors show that Them1 localization varies between being diffusely cytosolic to punctate-like condensates localizing near lipid droplets and mitochondria. They claim that Them1 localization is regulated by thermogenesis-signaling culminating in PKC β -dependent phosphorylation of Them1. Accordingly, phosphorylated Them1 is diffuse and cannot block lipid oxidation while hypophosphorylated Them1 is punctate and hinders β -oxidation of fatty acids, read out by measurements of oxygen consumption rates.

The manuscript is rationally organized and that the quality of the light microscopy data is good. The subject matter is quite topical and will be of interest to a broad audience. Below we suggest some additional experiments that could strengthen the conclusions made by the authors.

Major points:

1- The authors claim that three sites Them1 (S15,18 and 25) are PKC β targets. This PKC β consensus is X-S/T-X-R/K (Blom et al., 2004). Only S18 is a canonical one. The authors show by cell biology approaches that this site is actually the least important whereas S15 and S25 are more (surprisingly S25 is less conserved according to Supp Fig.1). To claim that Them1 is a direct target of PKC β the authors should perform in vitro kinase assays.

2- The method used for quantification of the Them1 volumes is unclearly described in Methods section and thus confusing. For instance, applying the same method leads to opposite results. In Figure 1b and 1i, Them1 condenses into puncta upon LY treatment and its volume shrinks. However, in Figure 2a-c, Them1 becomes more diffuse after 4 hour injections but still Them1 volume shrinks. Perhaps the authors should change their analysis method. Taking a segmentation approach using Imaris would allow the measurement of puncta volumes and intensities to be normalized to the total cell intensity allowing one to take in account the cell to cell variation. Additionally, the authors should be more explicit in similarities/differences between in vivo and in vitro cell culture experiments.

Related point: Fig 2(a,b,d,f), the authors should make more uniform the way they present images to allow side to side comparisons. Presently, there are a lot of presentation variations: 3D reconstruction or only sliced image, nuclei showed as green and not blue spheres in panel d. The authors must clarify and homogenize this.

3- The authors don't show the Ad-AAA and Ad-DDD in the Figure 4f. This is essential to prove that these mutants are not subjected to regulation downstream of PMA.

The authors could complement their OCR results by measuring for instance the levels of triglycerides. Could the phospho-dependent regulation of Them1 activity be measured in vitro?

Related point: Figures 4 e-f are labelled in a confusing way. Can the authors please use unique shapes/colours for the different conditions?

4- The authors provide interesting CLEM data. The Them1 puncta appear heterogeneous that could suggest that these are liquid-like or gel-like compartments. However, the authors don't provide any data about the diffusion rate of Them1 in this puncta, they rather comment on the in silico analysis of the NTerm region proposing phospho-dependent intermolecular interactions. This is purely speculative. The authors should test this hypothesis in vitro or move this speculation to discussion.

Minor points:

1- In the introduction, the authors could explain that norepinephrine is used to mimic cold exposure.

2- In Fig.1c, they should also provide references to support the proposed signalling pathway until PKC β .

3- In the paragraph "Phosphorylation-dependent changes in Them1 localization after stimulation", stronger support of PKC β -dependent phosphorylation would be appreciated. Them1 phosphorylation could be assessed by western blot if possible or mass spectrometry.

4- For the part "Puncta are biomolecular condensates, or "membraneless organelles", by correlative light/electron microscopy.", the authors could provide better evidence that Them1 forms condensates. The use of 1,6-hexanediol could help the authors to discern between liquid-like or solid-like structures (Kroschwald et al., 2017, DOI: <https://doi.org/10.19185/matters>).

Typing errors:

- p6: Replace "Supplementary Fig.3" by "Supplementary Fig.4"

- p8: Modify "Preliminary experiments established that an MOI of 1:40 yielded Them1 expression that was qualitatively similar to the tissue expression level of the protein in BAT of cold-exposed mice (Supplementary Fig. 2b) with a transfection efficiency of approximately 45%."

- p10: Replace "Supplementary Fig.4b" by "Supplementary Fig.5b" and "Supplementary Fig.4c,d" by "Supplementary Fig.5c,d"

- p10: Replace "Consistent with fluorescence micrographs" and "Consistent with fluorescence images"

- On the Fig.4c,d, add on the ordinate axis' titles GFP tagged and living cells counts/well, respectively

- In the legend of the Fig.1C, describe DAG, NE, β 3-AR

- In Methods, provide the magnification (high and low) used for the TEM acquisitions.

- In Methods, provide the treatments' duration in "Treatment of iBAs cells to explore metabolic pathways".

Reviewer #3 (Remarks to the Author):

General Comments

This paper concerns new insights into the mechanism through which Thioesterase Superfamily Member 1 (Them1) regulates thermogenesis, using cultured adipocytes and brown adipose tissue.

Although Them1 expression increases with cold exposure, it has previously been shown to reduce thermogenesis by inhibiting mitochondrial β -oxidation. To explain this apparent paradox, the authors now present evidence that stimuli which activate thermogenesis, such as a β_3 -Adr agonist, promote PKC β -dependent phosphorylation of Them1 at three identified sites. This in turn causes a change in cellular location, from lipid droplet- and mitochondrially-associated puncta to a diffuse cytosolic distribution. Them1 phosphorylation site mutants were shown to affect distribution and also cellular oxygen consumption, used as a surrogate measure of fatty acid oxidation. Finally the structural nature of the puncta was investigated, leading to the suggestion that unphosphorylated Them1 resides in membraneless organelles where it is active in hydrolysing fatty acylCoAs, preventing them entering mitochondria for β -oxidation.

Overall the findings are novel and will be of significant interest to others in the field. In addition to providing a mechanistic explanation for preventing the inhibitory effects of Them1 by thermogenic stimuli, the authors have identified a potential new level of fine-tuning and highlight the protein as a target for intervention.

Specific comments

1. The authors use *in silico* analysis to predict that Them1 could be phosphorylated by PKC β . As they point out, this kinase has been implicated in limiting energy expenditure and promoting obesity. PKC β deficiency increases fatty acid oxidation and reduces fat storage. However, this is not consistent with the data presented here that Them1 phosphorylation by PKC β causes redistribution of the protein and a higher OCR/ β -oxidation, which would increase energy expenditure.

2. The evidence that PKC β mediates the phosphorylation of Them1 in intact cells is based on the use of a phorbol ester, which is a non-specific activator of PKC isoforms, and the inhibitor LY333531/Ruboxistaurin, which is claimed to be specific for PKC β . However, this compound potentially inhibits several other kinases including PKC α , RSK1/2, PDK1 and PIM1/2/3 (Bain et al., *Biochem J*, 2007).

In silico analysis of Them1 using KinomeXplorer (Ref 13) or Netphos (Ref 14), as by the authors, does not list PKC β as the only, or even most likely, kinase to phosphorylate these sites, but also implicates PKC α and RSK among several others. Analysis by Scansite 4.0 (<https://scansite4.mit.edu/4.0/#home>) similarly highlights potential phosphorylation by several kinases, but only at low stringency analysis.

Better evidence for the role of PKC β needs to be provided. It will be difficult to knock down the kinase by siRNA given the low transfection efficiency reported. The authors may be able to see effects of a cell-permeant PKC β inhibitory peptide that acts independently of catalytic activity (e.g. Qvit N & Mochly-Rosen D, *Drug Discov Today Dis Mech*, 2010). In addition, PKC β activation may be observed by translocation assay or immunoblotting with phospho-specific antibodies under conditions leading to Them1 phosphorylation and loss of puncta. Can PKC β directly phosphorylate Them1 *in vitro*?

The authors will also need to address, at least in the Discussion, how PKC β activation downstream of NE or CL316,243, which will reduce the inhibition of β -oxidation, relates to the previously-described role of the kinase to limit energy expenditure and to promote lipid accumulation.

However, I believe identification of the kinase responsible for direct phosphorylation of these sites is not critical to the novelty and impact of the manuscript. This lies in the demonstration of changes in cellular localization in response to phosphorylation, and the resulting change in β -oxidation rate.

2. Fig. 1e indicates that it takes 4 h for PMA to induce full redistribution of Them1 in iBAs cells. Please show untreated control cells at 4 h.

3. The data presented in Fig 2 concern Them1 redistribution in CL316,243-injected mice, but are very confusingly analysed and described. The authors show a time-dependent effect of saline injection which they contribute to stress and temperature changes. The reference cited in support (Ref 18, "in press") is not available at this stage. In addition, the comparisons of 4 h CL316,243 with 1h saline data are therefore not appropriate and complicate the description of these results. Firstly, this section could be simplified by moving time-dependent changes in saline controls to a supplementary figure

and comparing only 4 h saline v CL316,243 data (i.e. Fig 2b vs 2f) in Fig. 2e.

Comparing the images presented in Fig 2b vs 2f clearly indicates changes in perilipin 1 organization and thus loss of lipid droplets, but the claimed differences in Them1 puncta volume, number and size at 4h are not very apparent.

The quantitative data in Fig. 2e showing an increase in Them1 volume in response to 4h CL316,243 are in agreement with the data from NE-treated cells in Fig. 1. However, the number of puncta increase >4 fold (compared with 4h saline) while the size decreases approx 2-fold. The average intensity does not differ between CL316,243 and saline treatment at 4 h. Thus overall the quantification suggests that CL316,243 actually increases the levels of Them1 residing in puncta.

4. The effect of WT and phosphorylation site mutants of Them1 on protein distribution and thus on acylCoA hydrolysis and fatty acid oxidation is the most important claim. The effect of serine mutation on Them1 localisation, assessed using plasmids or viral vectors, is extensively tested and convincing. The effect of the mutants on fatty acid oxidation is addressed indirectly by measuring OCR and needs to be strengthened. Can the authors show reciprocal effects of AAA and DDD mutants on oxidation of ¹⁴C-palmitate to ¹⁴CO₂? Ideally, the effects of all 4 constructs should be assessed ± PMA, at least on OCR.

Minor points

Pg. 4. "Them1 is comprised of tandem N-terminal thioesterase domains and a C-terminal lipid-binding domain steroidogenic acute regulatory-related lipid transfer (START) domain.

Delete "domain"

Pg. 4. The second 2/3s of the final paragraph in the Introduction give a lengthy description of the results, which could be shortened significantly.

Pg. 5. "...S18 is also conserved but less so because the serine residue can be substituted with alanine (Supplementary Fig. 1)."

Supplementary Fig. 1 does not give any indication of Ser to Ala substitution.

Pg. 6, end of last paragraph. The cycloheximide data are shown in Supplementary Fig. 4 not Supplementary Fig. 3. Also please correct spelling of cycloheximide throughout.

Pg. 7. Please clarify what is meant by "high Them1 at baseline" (line 6) and "Them1 localization was strong" (line 14). Please give scale bar size for Fig 2d, and the individual scale bar sizes for the 3 panels in Fig. 2f (not all 10 µm.)

Fig. 3a is confusing as it suggests that Thio-1/2 and START domains are deleted in the 3 mutants.

Pg. 9. "...whereas no effects are observed using Ad-EGFP controls (Fig. 4f)." A significant effect of PMA is indicated at 80 min.

Pg. 9 The description concerning Fig. 4e and 4f is omits important information. The text suggests that only 4f involved the additional effect of PMA preincubation before NE stimulation. However, the legend to Fig 4e states that these cells were also treated with PMA 30 min before NE stimulation.

Pg. 10. References made to Supplementary Fig. 4 should be to Supplementary Fig. 5.

Pg. 11. "...we also performed a motif scan on the first 20 amino acids. This analysis suggested a weak association..." There is no presentation of this analysis in Fig. 6 or Supplementary Fig. 5.

Pg. 14. "amino acids 20-45 represents a highly disordered area"

Pg. 14. "our data strongly supports"

Pg. 14. "...is the most likely area to investigate in the Them1 structure."

The authors of "**Thioesterase Superfamily Member 1 Undergoes Stimulus-coupled Reorganization to Regulate Metabolism**" **NCOMMS-20-10146-T** thank the reviewers for the constructive review of our originally submitted manuscript. The comments from all three reviewers were very helpful in helping us to revise the manuscript and the changes made the work far better-more rigorous and complete. We are thus grateful to the reviewers for their comments, which we addressed in the revised manuscript as follows (all text changes are in red font):

Reviewer 1

Reviewer #1 (Remarks to the Author):

Review Li et al.

The manuscript by Li et al. describes the importance of Them1 phosphorylation in the regulation of thermogenesis. Them1, a BAT-enriched long chain fatty acyl-CoA thioesterase, is upregulated by cold exposure and reduces fatty acid availability for β -oxidation in mitochondria to limit thermogenesis. However, the detailed molecular mechanisms of Them1 regulation remain unclear. The author showed that Them1-GFP localized in puncta nearby LD and mitochondria and that β -adrenergic stimulation diffuses Them1 into cytoplasm, likely because phosphorylation of Them1 by PKC β .

Although potentially of interest, the physiological significance of Them1 phosphorylation with respect to thermogenic regulation in vivo is largely unclear. Furthermore, definitive evidence of a change in phosphorylation of Them1 during thermogenic signaling is lacking. The comments below address some of the specific weaknesses of the present study.

Major points:

1. The authors conclude that adrenergic stimulation induces serine phosphorylation of Them1 based on only Ala or Asp substitution and public database. This claim is vital to the manuscript. However, no definitive evidence of change in Them1 phosphorylation by adrenergic stimulation is provided. The authors should perform phosphorylation assay (i.e. $^{32}\text{P}_i$ radiolabeling followed by pull down assay) to prove the dynamics of Them1 phosphorylation during adrenergic stimulation in iBA by comparison of Them1(WT) and Them1(AAA), at least.

Thank you for this important comment. To address this concern, we recruited Dr. Blaine Roberts from Emory University to perform a phosphoproteomic LSMS analysis of iBAs transduced with Ad-Them1-EGFP. For this, we stimulated iBAs with PMA, which results in the diffusion of Them1 from puncta (Fig. 1h), and looked specifically at Them1 phosphorylation events at the N-terminus from 5 min to 4 h after stimulation. The new results are in Fig. 1a, which show a significant time-dependent linear increase in phosphorylation events within the N-terminus of Them1 after stimulation. Results from the LSMS analysis also determined that phosphorylation events after PMA stimulation occurred at residues S25/S27, and S31 of Them1, shown in Fig. 1c. This was not identical to the human phosphomouse database, which indicated S15 phosphorylation occurs in vivo. We provide evidence in our mutational analysis (in Fig. 3) to support S15 phosphorylation and its important role in puncta formation/dissolution

in vitro. For a number of potential reasons, the LSMS results did not reveal phosphorylation events on S15 in iBAs, including S15 phosphorylation that could have been below detection in our study. These are enumerated in the revised discussion section, p 14, para 1. It would also be important to understand the role of S27 and S31 in the process. We plan to evaluate these scenarios in future studies.

2. Throughout the manuscript, the authors use EGFP-fusion Them1. They concluded that aggregation of Them1 was not due to EGFP, as evidenced by the diffused distribution of EGFP itself (Supplementary Figure 3a). However, EGFP forms a dimer at high concentration. They cannot rule out the possibility that localization of Them1 in certain region/organelle led to the formation of artificial aggregates of Them1-EGFP. In order to avoid artificial effect of tag on Them1 distribution, the authors should perform the same experiment in Figure 1a, b, d, f and Figure 3d, g on exogenously expressed untagged Them1 stained by anti-Them1 antibody they used in Figure 3 and Supplementary Figure 3b. The author should also perform EM study in Figure 5 using Them1-APEX2 without EGFP.

The reviewer's point is well taken. When setting-up the original experiments, we were mindful that aggregation of EGFP can occur. Because aggregation is dependent on pH and on the concentration of protein, we worked with low levels of intracellular Them1-EGFP (see Immunoblots, Supplementary Fig. 2a) and we made sure that the pH of cells in the culture experiments were maintained at a constant pH in buffered media. We felt that if the EGFP probe was homing to organelles or there were subdomains of the cytoplasm permissive to EGFP aggregation in the unstimulated state, where puncta (aggregates) are present, we would see puncta when using constructs containing Them1-EGFP or EGFP alone. This was not the case, as evidenced by our data in Supplementary Fig. 3a.

To address the reviewer's specific concern, we have now made a Them1 construct with a STOP codon to block translation of both EGFP and APEX2 and then stained iBAs with anti-Them1 antibody. The results can be found in Supplementary Fig. 3c, which verifies that puncta are present in unstimulated iBAs without the EGFP tag. Our EM study had (has) a sole aim to evaluate the high resolution structure of puncta, which helped immensely in identifying them as biomolecular condensates. In the EM study, EGFP/DAB was not found in lysosomes, mitochondria, or any other membrane-bound organelle. Instead, it was localized to membrane-less structures as described in Fig.5. We were able to do more validation experiments in the cultured cell system with a far greater *n*, so we did not repeat the cultured cell experiments using the Them1-STOP vector by EM. Additionally, without APEX2 we would have been unable to "find" the cells expressing Them1 and doing immunoEM for all of the conditions would have been beyond the scope of this investigation.

3. The sentence “~ Them1 was phosphorylated by PKC β at S15, ~” in abstract (page 2, line 10-11) and the scheme in Figure 3h are too assertive. The authors used PKC β inhibitor LY333531 to substantiate the putative involvement of PKC β . However, they failed to show Them1 distribution under PKC β knockout or knockdown condition. Furthermore, they didn't perform any in vitro kinase assay to show PKC β -dependent direct phosphorylation of Them1. If these experiments are too technically difficult, the authors should refrain from strong conclusions about the involvement of PKC β in Them1 translocation.

We fully agree with Reviewer 1. All reference to PKC- β phosphorylation of Them1 has now been removed. Please see Reviewer 2, query #1, for further discussion about the role of PKC β in our study.

4. The claims of PKC β involvement contradict the literature. According to the authors' hypothesis, PKC β -depletion promotes Them1-punctate formation and suppress β -oxidation. However, PKC β -/- mice exhibit increased energy expenditure as a result of increased mitochondrial fatty acid oxidation in brown adipose tissue (Bansode RP et al., JBC. 2007, PMID: 17962198 and Huang W et al., JLR. 2012, PMID: 22210924). How do the authors explain this discrepancy with their results? They should discuss this issue.

We thank the reviewer for raising this point, which was not discussed in our original submission. Indeed the overarching phenotype PKC β -/- mice is increased energy expenditure. The developmental absence of PKC β increase rates of FA oxidation, with associated and presumably adaptive increases in mitochondrial size and number in BAT and WAT (PMID: 17962198), as well as remodeling of WAT to become more BAT-like in the setting of leptin deficiency (PMID: 22210924). These adaptive phenotypes are ostensibly at odds with our findings that the activity of PKC β contributes to redistributing Them1 away from its active punctate configuration that suppresses fatty acid oxidation. However, our experiments have been conducted under conditions of short-term physiologic stimulation that are designed to test a hypothesis that transient Them1 re-distribution out of puncta reduce rates of fatty acid oxidation. We believe that our finding that PKC β signaling is among the stimuli that mediate short-term Them1 re-distribution is consistent with its role as a dynamic counter-regulatory mechanism that fine-tunes energy expenditure and enables energy conservation when the immediate stimulus for heat production in BAT is removed. We have modified the discussion section to address this issue, taking into account revisions that further qualify the role of PKC β in Them1 redistribution.

5. The data shown in Figure 2 are crucial to show the translocation of Them1 in vivo. However, the possibility of non-specific staining cannot be excluded. Thus, the author should show the specificity of anti-Them1 antibody using BAT from Them1 KO mice. Furthermore, the results shown in Figure 2 show the mixed effect of both increased body temperature and CL injection. The authors acclimate mice to thermoneutral or optimized temperature to stabilize Them1 expression before CL316243 injection.

The Them1 antibody was made by us and had been fully characterized using WT and KO mice (PMID: 22993230; PMID: 27110486). We have been using the anti-Them1 antibody for immunostaining studies since our original publication in 2012. After completing studies for the current manuscript, we unfortunately depleted our affinity purified antibody stock and had to make more to complete the revision. For this, we made the same synthetic peptide to Them1 and affinity purified our remaining stock of serum from the original Them1 antibody production run. Because there were 2 original bleeds, we chose the 2nd bleed because it was in highest concentration after purification. We then fully characterized this new lot of antibody as had been done originally. We added new immunoblot data in Supplementary Fig. 4a and new immunostaining data with wild-type and Them1-deficient mice to demonstrate specificity of the antibody in Supplementary Figure 4b.

The way in which methods were original written for this section was unclear and confusing. This has been corrected in the revised manuscript, page 24, para 2 and 3. In brief, the mice were kept in mild hypothermic conditions (22^o C) after arrival from the vendor and then 24 h after being moved to the metabolic cages. They were then equilibrated to normothermic conditions overnight prior to the

experiment and kept at normothermic temperature throughout the experiment. The data thus do not account for both temperature and stimulation, but just for the effects of stimulation.

6. The title of this manuscript is too vague. It should be added that this study focused on brown adipocytes.

We appreciate that all of the genetic mutation work had to be done in cultured cells, but we have many years of in vivo data concerning the metabolic regulation with Them1 and feel that the in vitro data help us to understand the in vivo data that are already published in WT and KO mice. We have also validated our in vitro studies using mice in this study. We thus appeal to Reviewer 1 that the title as stands addresses important issues in in vivo BAT rather than just being about cultured cells per se.

Minor points:

1. In Figure 1a, b, the authors should also stain other organelles such as lysosome, peroxisome, Golgi apparatus and ER.

Historically, Them1 is has been demonstrate to regulate metabolism by interacting with mitochondria and lipid droplets. Although it would be interesting to stain the other organelles listed, we have no evidence that Them1 localizes to these organelles, so we were not certain the value of staining for them here.

2. Figure 3c-g needs statistics. The authors need to measure Them1-punctate volume in multiple cells, to discuss the contribution of each serine residue to Them1 distribution.

The reviewer's point is well taken. Quantification of the volume of Them1 in the S15-S18-S25D (DDD) mutant and the S15D, S18D, and S25D Them1 mutants have now been added to Figure 3 as Fig. 3g.

3. Page 6, line 21; “~concentration (Supplemental Fig 3)” should be “~ concentration (Supplemental Fig 4)”

In the revised manuscript, these data can be found in Supplementary Fig. 6. The corresponding change was made in the revised text page 7, line 3.

4. Page 10, line 5 and line 9; The author need to be clear which Them1 mutant was used (i.e. ~ puncta in Them1(AAA)-EGFP-APEX2 cells were ~).

Thank you for bringing this to our attention. We did not realize that this section was so unclear. We added text to ensure the vectors used are clearly stated. Changes made in the revised text are on pages 9, para 2 and 10 para 1.

Reviewer 2

Reviewer #2 (Remarks to the Author):

Comments of the reviewers: In this manuscript, Yue Li et al. investigate the role the Thioesterase Superfamily Member 1 (Them1) plays in thermogenesis in brown adipose tissue. The authors show that Them1 localization varies between being diffusely cytosolic to punctate-like condensates localizing near lipid droplets and mitochondria. They claim that Them1 localization is regulated by thermogenesis-signaling culminating in PKC β -dependent phosphorylation of Them1. Accordingly, phosphorylated Them1 is diffuse and cannot block lipid oxidation while hypophosphorylated Them1 is punctate and hinders β -oxidation of fatty acids, read out by measurements of oxygen consumption rates. The manuscript is rationally organized and that the quality of the light microscopy data is good. The subject matter is quite topical and will be of interest to a broad audience. Below we suggest some additional experiments that could strengthen the conclusions made by the authors.

Major points:

1- The authors claim that three sites Them1 (S15,18 and 25) are PKC β targets. This PKC β consensus is X-S/T-X-R/K (Blom et al., 2004). Only S18 is a canonical one. The authors show by cell biology approaches that this site is actually the least important whereas S15 and S25 are more (surprisingly S25 is less conserved according to Supp Fig.1). To claim that Them1 is a direct target of PKC β the authors should perform in vitro kinase assays.

Thank you for the important suggestion. We agree with the reviewer that the sites do not seem to be PKC β target consensus sites. To verify this experimentally, we used the CST phospho-(Ser) PKC substrate antibody, which recognizes endogenous levels of cellular proteins only when phosphorylated at serine residues surrounded by R/K at the -2 and +2 positions and a hydrophobic residue at the +1 position. The antibody did not cross-react with nonphosphorylated serine residues, with phosphothreonine in the same motif, or with phospho-serine in other motifs. The data are provided in Supplementary Fig. 7, which show virtually no immunostaining in blots of Them1 purified by IP using an EGFP pull-down. Interestingly, the same data do show that a number of proteins (in the input) were phosphorylated by PKC, and these may be Them1 binding partners. We appreciate the insightful comment, which facilitated the identification of a number of protein targets to pursue that may regulate puncta associations/disassociations after stimulation. For the revised manuscript, we deleted all reference to PKC β 's direct role in phosphorylating Them1 and regulating Them1 function. We left-open the possibility, however, that PKC's role is indirect by regulating phosphorylation events on Them1 binding partners or other interacting proteins that regulate puncta formation and/or dissolution. This is illustrated in Fig. 1f, where we used a dashed line to denote PKC's putative role(s) in regulating Them1 (Figure legends, Fig. 1f, page 35). We also added a statement about the binding specificity of the CST antibody on page 7, para 1.

2- The method used for quantification of the Them1 volumes is unclearly described in Methods section and thus confusing. For instance, applying the same method leads to opposite results. In Figure 1b and 1i, Them1 condenses into puncta upon LY treatment and its volume shrinks. However, in Figure 2a-c, Them1 becomes more diffuse after 4 hour injections but still Them1 volume shrinks. Perhaps the authors should change their analysis method. Taking a segmentation approach using Imaris would allow the measurement of puncta volumes and intensities to be normalized to the total cell intensity allowing

one to take in account the cell to cell variation. Additionally, the authors should be more explicit in similarities/differences between in vivo and in vitro cell culture experiments.

Related point: Fig 2(a,b,d,f), the authors should make more uniform the way they present images to allow side to side comparisons. Presently, there are a lot of presentation variations: 3D reconstruction or only sliced image, nuclei showed as green and not blue spheres in panel d. The authors must clarify and homogenize this.

Thank you for this suggestion. We have now changed our experimental design for quantifying images in Figure 2. For this, rather than using the Zeiss Airyscan (near super-resolution) to acquire images in Figure 2, we re-did all of the images using the confocal microscope. Confocal images in the revised manuscript replaced the Airyscan images originally submitted as Fig. 2 d-f. This standardized the input data the analysis of volumes in Figures 1, 2, and 3.

The reviewer's points are well taken and suggestions appreciated. For experiments aimed to quantify changes in intercellular Them1 localization after stimulation, we used image segmentation, as recommended, to calculate the volume of Them1 by thresholding the green (EGFP) pixels of specified intensity that represented Them1-EGFP minus background intensity. Our Volocity software license expired and we opted not to renew but instead purchased new software from Imaris. To ensure that all of the data was analyzed in a similar manner, we re-calculated all of the volume measurements from the original data in Fig. 1k and Fig. 3g (revised manuscript), and then did volume measurements for the newly acquired images in Fig. 2. We changed the text in the Methods section accordingly and have also better explained the way we did the in vitro vs in vivo analysis on page 24, para 4 and page 25, para 1.

We have now changed the format of the images in Fig. 2 to be more consistent, although we use the 3D reconstruction for an important reason of being able to visualize the localization of Them1 in vivo with more depth. For the remainder of the images, we made the panels consistent, with the Them1 image (green) left and the merged image RGB image to the right.

One important result we show in Fig. 2 is that Them1 translocates into the nucleus after stimulation. This is why the nuclei, particularly at 4 h after CL exposure, are labeled green (i.e. because they express Them1-EGFP). Adding the blue (Hoechst, HOE) assists with delineating the structures are nuclei, as requested, but we cannot remove the green because it represents data.

3- The authors don't show the Ad-AAA and Ad-DDD in the Figure 4f. This is essential to prove that these mutants are not subjected to regulation downstream of PMA.

The authors could complement their OCR results by measuring for instance the levels of triglycerides. Could the phospho-dependent regulation of Them1 activity be measured in vitro?

Related point: Figures 4 e-f are labelled in a confusing way. Can the authors please use unique shapes/colours for the different conditions?

Experiments with Ad-AAA and Ad-DDD have been added to Fig. 4 as Fig. 4f. They show that these mutants are not subjected to regulation downstream of PMA.

Levels of triglycerides have been added as Supplementary Fig. 8c.

Experiments to show the phosphorylation-dependent regulation of Them1 activity have been added as Supplementary Fig. 8d.

The labeling was changed so that each different condition and construct has been assigned a specific color and/or line shape (see Fig. 4c).

4- The authors provide interesting CLEM data. The Them1 puncta appear heterogeneous that could suggest that these are liquid-like or gel-like compartments. However, the authors don't provide any data about the diffusion rate of Them1 in this puncta, they rather comment on the in silico analysis of the NTerm region proposing phospho-dependent intermolecular interactions. This is purely speculative. The authors should test this hypothesis in vitro or move this speculation to discussion.

The reviewer's point is well taken. Our in silico analyses are consistent with the hypothesis that the ultrastructural images represent membraneless organelles. Experimental testing of Them1 phase transitions will certainly be a central goal of our future studies. As suggested by the reviewer for the current manuscript, we have limited our speculations to the discussion section.

Minor points:

1- In the introduction, the authors could explain that norepinephrine is used to mimic cold exposure.

This has been added to page 4, para 2.

2- In Fig.1c, they should also provide references to support the proposed signalling pathway until PKC β .

Thank you for the suggestion, but the experiments in this paper have established this pathway.

3- In the paragraph "Phosphorylation-dependent changes in Them1 localization after stimulation", stronger support of PKC β -dependent phosphorylation would be appreciated. Them1 phosphorylation could be assessed by western blot if possible or mass spectrometry.

Thank you for the comment. We added LSMS work to show that Them1 phosphorylation occurs is in Figure 1a and c.

4- For the part "Puncta are biomolecular condensates, or "membraneless organelles", by correlative light/electron microscopy.", the authors could provide better evidence that Them1 forms condensates. The use of 1,6-hexanediol could help the authors to discern between liquid-like or solid-like structures (Kroschwald et al., 2017, DOI: <https://doi.org/10.19185/matters>).

Thank you for the suggestion. Please see the response to query #4. In summary: The reviewer's point is well taken. Our in silico analyses are consistent with the hypothesis that the ultrastructural images represent membraneless organelles. Experimental testing of Them1 phase transitions will certainly be a central goal of our future studies.

Typing errors: - p6: Replace “Supplementary Fig.3” by “Supplementary Fig.4”

- p8: Modify “Preliminary experiments established that an MOI of 1:40 yielded Them1 expression that was qualitatively similar to the tissue expression level of the protein in BAT of cold-exposed mice (Supplementary Fig. 2b) with a transfection efficiency of approximately 45%.”

- p10: Replace “Supplementary Fig.4b” by “Supplementary Fig.5b” and “Supplementary Fig.4c,d” by “Supplementary Fig.5c,d”

- p10: Replace “Consistent with fluorescence micrographs” and “Consistent with fluorescence images”

- On the Fig.4c,d, add on the ordinate axis' titles GFP tagged and living cells counts/well, respectively

- In the legend of the Fig.1C, describe DAG, NE, β 3-AR

- In Methods, provide the magnification (high and low) used for the TEM acquisitions.

- In Methods, provide the treatments' duration in “Treatment of iBAs cells to explore metabolic pathways”.

“Supplementary Fig.3” was replaced by “Supplementary Fig.6” on page 7, para 1.

“Preliminary experiments established that an MOI of 1:40 yielded Them1 expression that was qualitatively similar to the tissue expression level of the protein in BAT of cold-exposed mice (Supplementary Fig. 2b) with a transfection efficiency of approximately 45%” was modified to “Them1 expression at an MOI of 1:40 was approximately 45%, which resulted in Them1 expression similar to tissue expression in BAT of cold-exposed mice (Supplementary Fig. 2b)” on p9, para 1.

“Supplementary Fig.4b” was replaced by “Supplementary Fig.9b” on p11, para 2 and “Supplementary Fig.4c,d” by “Supplementary Fig.9c,d” on p11, para 2

“Consistent with fluorescence micrographs” was replaced with “Consistent with fluorescence images” on p11, para 1.

Fig.4c,d from the original manuscript were deleted in the revised manuscript.

The legend to Fig.1C was modified to read Abbreviations: β ₃-AR, beta-3 adrenergic receptor; DAG, diacylglycerol; NE, norepinephrine” on page 35.

Low magnification TEM images were taken at an original magnification of 4,000-5,000x, intermediate magnification TEM images at 25,000x, and high magnification TEM images at 60,000x. This information was added to Methods, p22, para 2 and to the figure legends.

In the Method section, the following changes were made to the section “Treatment of iBAs cells to explore metabolic pathways” to indicate the timing of the addition of stimulators and inhibitors on p21 para 3.

“To explore the metabolic pathway leading to re-organization of intracellular Them1, various treatments were applied for 4 h to cultured iBAs and on p22, para 1.

“Pathway inhibitors were added 1 h prior to activation and were continued throughout the 4 h activation period as follows:

Although not communicated in the paper, per se, we had also done cell viability experiments in each condition to ensure none of the treatments/time reduced cell viability.

Reviewer 3

Reviewer #3 (Remarks to the Author):

General Comments This paper concerns new insights into the mechanism through which Thioesterase Superfamily Member 1 (Them1) regulates thermogenesis, using cultured adipocytes and brown adipose tissue. Although Them1 expression increases with cold exposure, it has previously been shown to reduce thermogenesis by inhibiting mitochondrial β -oxidation. To explain this apparent paradox, the authors now present evidence that stimuli which activate thermogenesis, such as a β_3 -Adr agonist, promote PKC β -dependent phosphorylation of Them1 at three identified sites. This in turn causes a change in cellular location, from lipid droplet- and mitochondrially-associated puncta to a diffuse cytosolic distribution. Them1 phosphorylation site mutants were shown to affect distribution and also cellular oxygen consumption, used as a surrogate measure of fatty acid oxidation. Finally the structural nature of the puncta was investigated, leading to the suggestion that unphosphorylated Them1 resides in membraneless organelles where it is active in hydrolysing fatty acylCoAs, preventing them entering mitochondria for β -oxidation. Overall the findings are novel and will be of significant interest to others in the field. In addition to providing a mechanistic explanation for preventing the inhibitory effects of Them1 by thermogenic stimuli, the authors have identified a potential new level of finetuning and highlight the protein as a target for intervention.

1. The authors use in silico analysis to predict that Them1 could be phosphorylated by PKC β , As they point out, this kinase has been implicated in limiting energy expenditure and promoting obesity. PKC β deficiency increases fatty acid oxidation and reduces fat storage. However, this is not consistent with the data presented here that Them1 phosphorylation by PKC β causes redistribution of the protein and a higher OCR/ β -oxidation, which would increase energy expenditure.

The reviewer's point is well taken and expresses the same concern as reviewer 1 (point 4). Our response to both reviewers is as follows:

We thank the reviewer for raising this point, which was not discussed in our original submission. Indeed the overarching phenotype PKC β ^{-/-} mice is increased energy expenditure. The developmental absence of PKC β increase rates of FA oxidation, with associated and presumably adaptive increases in mitochondrial size and number in BAT and WAT (PMID: 17962198), as well as remodeling of WAT to become more BAT-like in the setting of leptin deficiency (PMID: 22210924). These adaptive phenotypes are ostensibly at odds with our findings that the activity of PKC β contributes to redistributing Them1

away from its active punctate configuration that suppresses fatty acid oxidation. However, our experiments have been conducted under conditions of short-term physiologic stimulation that are designed to test a hypothesis that transient Them1 re-distribution out of puncta reduce rates of fatty acid oxidation. We believe that our finding that PKC β signaling is among the stimuli that mediate short-term Them1 re-distribution is consistent with its role as a dynamic counter-regulatory mechanism that fine-tunes energy expenditure and enables energy conservation when the immediate stimulus for heat production in BAT is removed. We have modified the discussion section to address this issue, taking into account revisions that further qualify the role of PKC β in Them1 redistribution.

2. The evidence that PKC β mediates the phosphorylation of Them1 in intact cells is based on the use of a phorbol ester, which is a non-specific activator of PKC isoforms, and the inhibitor LY333531/Ruboxistaurin, which is claimed to be specific for PKC β . However, this compound potentially inhibits several other kinases including PKC α , RSK1/2, PDK1 and PIM1/2/3 (Bain et al., Biochem J, 2007).

In silico analysis of Them1 using KinomeXplorer (Ref 13) or Netphos (Ref 14), as by the authors, does not list PKC β as the only, or even most likely, kinase to phosphorylate these sites, but also implicates PKC α and RSK among several others. Analysis by Scansite 4.0 (<https://scansite4.mit.edu/4.0/#home>) similarly highlights potential phosphorylation by several kinases, but only at low stringency analysis.

Better evidence for the role of PKC β needs to be provided. It will be difficult to knock down the kinase by siRNA given the low transfection efficiency reported. The authors may be able to see effects of a cell-permeant PKC β inhibitory peptide that acts independently of catalytic activity (e.g. Qvit N & Mochly-Rosen D, Drug Discov Today Dis Mech, 2010). In addition, PKC β activation may be observed by translocation assay or immunoblotting with phosphospecific antibodies under conditions leading to Them1 phosphorylation and loss of puncta. Can PKC β directly phosphorylate Them1 in vitro?

The authors will also need to address, at least in the Discussion, how PKC β activation downstream of NE or CL316,243, which will reduce the inhibition of β -oxidation, relates to the previously-described role of the kinase to limit energy expenditure and to promote lipid accumulation.

However, I believe identification of the kinase responsible for direct phosphorylation of these sites is not critical to the novelty and impact of the manuscript. This lies in the demonstration of changes in cellular localization in response to phosphorylation, and the resulting change in β -oxidation rate.

We agree with reviewer 3 completely and are currently working to determine the kinase involved in Them1 phosphorylation. We found, in Supplementary Fig. 7 that Them1, per se, is not a PKC substrate. However, six or more proteins are phosphorylated on serine residues by PKC, which may be involved in puncta formation/dissolution. Once we identify these PKC substrate proteins, we can determine whether they are phosphorylated by PKC β .

To address how PKC β activation downstream of NE or CL316,243, which will reduce the inhibition of β -oxidation, relates to the previously-described role of the kinase to limit energy expenditure and to promote lipid accumulation, we revised the discussion on page 14, para 1.

We appreciate that reviewer 3 does not think it is critical to identify the kinase responsible for direct phosphorylation of Them1. This reviewer communicated that it is most important for the impact of our work to demonstrate changes in cellular localization in response to phosphorylation and the resulting change in β -oxidation rate. We have bolstered many of the observations originally made to provide an even more robust demonstration of these relationships.

2. Fig. 1e indicates that it takes 4 h for PMA to induce full redistribution of Them1 in iBAs cells. Please show untreated control cells at 4 h.

These new data are provided in Supplementary Fig. 5.

3. The data presented in Fig 2 concern Them1 redistribution in CL316,243-injected mice, but are very confusingly analysed and described. The authors show a time-dependent effect of saline injection which they contribute to stress and temperature changes. The reference cited in support (Ref 18, "in press") is not available at this stage. In addition, the comparisons of 4 h CL316,243 with 1h saline data are therefore not appropriate and complicate the description of these results. Firstly, this section could be simplified by moving time-dependent changes in saline controls to a supplementary figure and comparing only 4 h saline v CL316,243 data (i.e. Fig 2b vs 2f) in Fig. 2e. Comparing the images presented in Fig 2b vs 2f clearly indicates changes in perilipin 1 organization and thus loss of lipid droplets, but the claimed differences in Them1 puncta volume, number and size at 4h are not very apparent. The quantitative data in Fig. 2e showing an increase in Them1 volume in response to 4h CL316,243 are in agreement with the data from NE-treated cells in Fig. 1. However, the number of puncta increase >4 fold (compared with 4h saline) while the size decreases approx 2-fold. The average intensity does not differ between CL316,243 and saline treatment at 4 h. Thus overall the quantification suggests that CL316,243 actually increases the levels of Them1 residing in puncta.

We agree with Reviewer 3 and appreciate that this section was confusing. One issue we had was taking some images using the confocal, with lower resolution, and some with our brand new Airyscan attachment, which provided near super-resolution images. We rectified this by repeating all of the immunostaining and quantification of BAT tissues using conventional confocal images, so we were comparing apples to apples. We also purchased Imaris software, which allowed us to better segment the data in images to analyze them. We re-did the analysis of data in Figure 1k first, which validated this approach by providing the same data we had previously. We then re-did the analyses in Figure 2 using the new software and methods on conventional confocal images. Because the important objective was to show that the volume of Them1 increased significantly over time, we decided to present the data in a simple linear regression line, volume vs time, in Fig. 2c. We then replaced the previous images at 1, 2, and 4 h with new images taken with the confocal. We chose to still present the saline control cells, done with the Airyscan attachment, because it allows us to clearly demonstrate the localization of puncta, in vivo, which is important data. Because our endpoint with CL was 4 h, it was also important to show that puncta were present after saline injection at 4 h.

4. The effect of WT and phosphorylation site mutants of Them1 on protein distribution and thus on acylCoA hydrolysis and fatty acid oxidation is the most important claim. The effect of serine mutation on

Them1 localisation, assessed using plasmids or viral vectors, is extensively tested and convincing. The effect of the mutants on fatty acid oxidation is addressed indirectly by measuring OCR and needs to be strengthened. Can the authors show reciprocal effects of AAA and DDD mutants on oxidation of 14C-palmitate to 14CO₂? Ideally, the effects of all 4 constructs should be assessed ± PMA, at least on OCR.

We agree and have performed the requested new experiments. To Fig. 4, we added OCR values ± PMA and we added stimulated oxidation data for AAA vs DDD to Supplementary Fig. 8d. We revised the text in the Results and Methods sections accordingly.

Minor points:

Pg. 4. “Them1 is comprised of tandem N-terminal thioesterase domains and a C-terminal lipid-binding domain steroidogenic acute regulatory-related lipid transfer (START) domain. Delete “domain”

Domain was deleted as requested.

Pg. 4. The second 2/3s of the final paragraph in the Introduction give a lengthy description of the results, which could be shortened significantly.

We have shortened this section by deleting some text, page 4, second 2/3 of the final paragraph.

Pg. 5. “...S18 is also conserved but less so because the serine residue can be substituted with alanine (Supplementary Fig. 1).” Supplementary Fig. 1 does not give any indication of Ser to Ala substitution.

We agree, thank you. This has been deleted from the revised manuscript, page 5, para 1.

Pg. 6, end of last paragraph. The cycloheximide data are shown in Supplementary Fig. 4 not Supplementary Fig. 3. Also please correct spelling of cycloheximide throughout.

We regret these typos, which have been corrected in the revised manuscript.

Pg. 7. Please clarify what is meant by “high Them1 at baseline” (line 6) and “Them1 localization was strong” (line 14).

Them1 expression increases with decreasing ambient temperature, which can be seen in the immunoblots, Supplementary Fig. 2b and in ref. 11. Because of this, we housed mice at 22° C, which is below their thermoneutral zone and this intervention increases Them1 expression considerably. This is what we meant by the statement. We modified that text in the revised manuscript, page 7, para 3.

Please give scale bar size for Fig 2d, and the individual scale bar sizes for the 3 panels in Fig. 2f (not all 10 µm.)

We deleted Fig 2d. All paired panels in Fig. 2 have a scale bar in the revised manuscript.

Fig. 3a is confusing as it suggests that Thio-1/2 and START domains are deleted in the 3 mutants.

We have revised the construct figure in Figure 3a.

Pg. 9. "...whereas no effects are observed using Ad-EGFP controls (Fig. 4f)." A significant effect of PMA is indicated at 80 min.

This sentence was deleted in the revised manuscript.

Pg. 9 The description concerning Fig. 4e and 4f is omits important information. The text suggests that only 4f involved the additional effect of PMA preincubation before NE stimulation. However, the legend to Fig 4e states that these cells were also treated with PMA 30 min before NE stimulation.

The Figure and methodological details have been edited.

Pg. 10. References made to Supplementary Fig. 4 should be to Supplementary Fig. 5.

The text was edited.

Pg. 11. "...we also performed a motif scan on the first 20 amino acids. This analysis suggested a weak association..." There is no presentation of this analysis in Fig. 6 or Supplementary Fig. 5.

Because the association was weak and only observed when using the first 20 aa of Them1 in the query, we decided it was prudent to change this statement. When using the entire Them1 sequence in the query, rather than just the first 20 aa, there is no association of the N-terminus with other proteins. We changed the statement to read there are no strong interactions with the N-terminus that are denoted using this query site. The revised statement is included on p 12 para 1.

Pg. 14. "amino acids 20-45 represents a highly disordered area"

Pg. 14. "our data strongly supports"

The text was edited as requested on p 15 of the revised manuscript.

Pg. 14. "...is the most likely area to investigate in the Them1 structure."

We deleted that statement on p15 of the revised manuscript and added the following: "our data strongly support that this region is involved in puncta formation and should be further investigated."

REVIEWERS' COMMENTS

Reviewer #1 (Remarks to the Author):

The authors have provided a reasonable response to the concerns raised.

Reviewer #2 (Remarks to the Author):

The authors have more-or-less addressed my previous concerns.

However, my request to assess if Ad-AAA and Ad-DDD are subject to regulation downstream of PMA was not properly addressed. The point was not to directly compare Ad-AAA and Ad-DDD (current Figure 4F) but to compare Ad-AAA + PMA to Ad-AAA – PMA (and the same for Ad-DDD), similarly to what is plotted in Figure 4d and 4e. I would request that these direct comparisons be provided.

Minor comments

The sentence on page 7 "Because the canonical PKC binding sequence, R/K-X-S-X-R/K where X at +1 is a hydrophobic residue, is not present within the N-terminal Them1 sequence (Fig. 1c), we used an antibody specific for this sequence in immunoblots of Them1 after stimulation with PMA over time (Supplementary Fig. 7) to explore the hypothesis that PKC phosphorylates Them1" sounds off (why probe for a sequence that is already known not to be present). Could the authors please reword this sentence.

Reviewer #3 (Remarks to the Author):

The authors have performed several new experiments and significantly revised the manuscript to address all of the reviewers' comments and should be commended for improving the study. Attention to the following minor issues is required:

The phosphoproteomic data for Them1 supports their argument but is presented as a normalized aggregate abundance of N-terminal phosphopeptides (Fig 1a). Why are the changes at the individual sites, most of which can apparently be measured specifically, not presented? The choice of a phosphopeptide from HSL as a reference for normalization should also be explained – what is the evidence that this site is not affected by PMA?

Typo: phosphorylated peptide RSS(phospho)QGVLMPLYTSPIVK....

The scheme presented in Fig.1f needs further modification. DAG released by ATGL is in the sn-1,3 or sn-2,3 stereochemical isoforms, so that TAG-derived DAG cannot activate protein kinase C (which requires sn-1,2 DAG). It is possible that some DAG released in this way may undergo isomerization, but this should be discussed.

Eichmann et al., (2012). Studies on the substrate and stereo/regioselectivity of adipose triglyceride lipase, hormone-sensitive lipase, and diacylglycerol-O-acyltransferases. *J Biol Chem* 287, 41446-41457.

The authors of "Thioesterase Superfamily Member 1 Undergoes Stimulus-coupled Conformational Reorganization to Regulate Metabolism in Mice" NCOMMS-20-10146-A thank the reviewers for the positive review and outcome of our revised manuscript. We address the remaining concerns in the revised A2 manuscript, which also includes numerous changes requested by the editor (summarized in the Editorial Requests document), as follows (all text changes are in red font):

Reviewer 1

Reviewer #1 (Remarks to the Author):

Comments of the reviewer: The authors have provided a reasonable response to the concerns raised.

Thank you; we are pleased to have responded to the reviewer's concerns adequately.

Reviewer 2

Reviewer #2 (Remarks to the Author):

Comments of the reviewer: The authors have more-or-less addressed my previous concerns. However, my request to assess if Ad-AAA and Ad-DDD are subject to regulation downstream of PMA was not properly addressed. The point was not to directly compare Ad-AAA and Ad-DDD (current Figure 4F) but to compare Ad-AAA + PMA to Ad-AAA – PMA (and the same for Ad-DDD), similarly to what is plotted in Figure 4d and 4e. I would request that these direct comparisons be provided.

Reviewer 2's point is well taken that the direct comparison between iBA cells transfected with Ad-AAA (and Ad-DDD) with and without PMA prior to NE stimulation is required to determine whether AAA and DDD forms of Them1 are subject to downstream regulation by PMA. We have now performed these experiments, and the data are incorporated into the manuscript. These results argue against downstream regulation of these protein constructs, but must be interpreted in the context of the effects of PMA *per se* on iBAs observed under the experimental conditions utilized to measure oxygen consumption rates (OCR) as a surrogate marker of fatty acid oxidation.

In contrast to the experiments in the original Fig. 4e in which PMA did not appreciably influence OCR in cells transduced with EGFP, in the current studies we observed that PMA suppressed OCR values in non-transduced iBAs, as well as in cells transduced with EGFP (see Fig. Ai-ii).

Figure A: iBAs transduced with Ad-EGFP (green) and non-transfected controls (black) were treated with vehicle or phorbol 12-myristate 13-acetate (PMA) 30 min before norepinephrine (NE) stimulation. Oxygen consumption rate (OCR) as % of baseline in (i) Original Figure 4e, and (ii) Current experiments. $n = 19-40$ (i) and $n = 33$ (ii), combined data from $n = 3$ independent experiments. Error bars indicate mean \pm SE.

The differences were most likely attributable to technical limitations of culturing and differentiating iBAs within the extremely small well sizes available in the 96-well plates of the Seahorse apparatus. These constraints can lead to differences in degrees of differentiation and confluence that may influence cellular metabolism, such that substrates can variably become limiting when presented

to cells in very small volumes of media.

Support for this is shown in **Fig. Bi-ii** (and

Supplementary Fig. 9c), which compares

extracellular acidification rates (ECAR, top

panels) and OCR values (bottom panels)

following the addition of PMA, prior to NE

exposure from the

experiments presented

in Fig. 4d and 4e of the original manuscript

compared with the

current experiments.

Baseline values of OCR

and ECAR (i.e. time 0),

which reflect culture conditions, were higher in the original compared with the current

experiments. Under the original conditions, the addition of PMA led to an initial suppression of

ECAR values that returned to baseline just prior to NE exposure. By contrast, in the current

experiments, addition of PMA led to a progressive increase in ECAR values until the time of NE

exposure. In both series of experiments, the differences in the response of ECAR to PMA

observed prior to NE addition, occurred in the absence of differences in OCR. This is indicative

of PMA stimulation of anaerobic glycolysis, as opposed to fatty acid oxidation, which increases

OCR following NE exposure. In this connection, PMA has been shown by others to stimulate

glycolysis in both neutrophils (Immunology 2015 Jun; 145(2): 213–224) and fibroblasts (PNAS

1985 Oct; 82:6440-6444). On this basis, we infer that PMA-induced increases in anaerobic

glycolysis in the current experiments led to the observed reductions in fatty acid oxidation, as

reflected by OCR in response to NE stimulation, expressed as a % of baseline (see **Fig. Ci**).

To account for the variable effects of PMA *per se* on cellular metabolism based on culture

conditions, we analyzed the data in the current experiments by quantifying the effects of NE for

each experimental condition as the area under the curve (AUC) following exposure. In order to

compare the effects of PMA on Ad-DDD, Ad-AAA and Ad-EGFP, we subtracted AUC values for

transduced cells in the presence of PMA from cells exposed to vehicle. *Our findings reveal that*

the NE-stimulated OCR of Ad-AAA iBAs was suppressed compared to Ad-DDD and Ad-EGFP

(see **Fig. Cii-iii** and Fig. 5e,f of the revised manuscript), and argue against downstream

Figure B: iBAs transduced with Ad-Them1-EGFP (blue), Ad-EGFP (green), Ad-AAA-EGFP (red) or Ad-DDD-EGFP (violet) and non-transfected controls (black) were treated with vehicle or phorbol 12-myristate 13-acetate (PMA) 30 min prior to norepinephrine (NE) stimulation. Extracellular acidification rate (ECAR; top panels) and oxygen consumption rate (OCR; bottom panels). Data correspond to that from (i) original Figures 4d,e, and (ii) current experiments. $n = 19-45$ (i) and $n = 33-36$ (ii), combined data from $n = 3$ independent experiments. Error bars indicate mean \pm SE.

Figure C: iBAs transduced with Ad-EGFP (green), Ad-AAA-EGFP (red) or Ad-DDD-EGFP (violet) and non-transfected controls (black) were treated with vehicle or phorbol 12-myristate 13-acetate (PMA) 30 min before norepinephrine (NE) stimulation. Oxygen consumption rate (OCR) as a % of baseline with all data (i) and from NE stimulation only (ii). Data were evaluated using mixed-effects analysis for Vehicle and PMA treated groups (Vehicle: Group effect $P=0.050$, Interaction effect $P<0.001$; PMA: Group effect $P=0.060$, Interaction effect $P=0.015$) with multiple comparison testing using the Holm-Sidak method (Vehicle: $^{\#}P=0.005$ and $^{\#}P=0.003$ Ad-EGFP vs. Ad-AAA-EGFP, $^*P=0.034$, Ad-DDD-EGFP vs. Ad-AAA-EGFP; PMA: $^{\#}P=0.032$, $^{\#}P=0.024$ and $^{\#}P=0.024$ Ad-EGFP vs. Ad-AAA-EGFP). (iii) Change in area under the curve (Δ AUC) values from baseline, Vehicle - PMA. Data were evaluated using a one-way ANOVA with multiple comparison testing using the Holm-Sidak method ($^*P=0.046$, Ad-DDD-EGFP vs. Ad-AAA-EGFP, $^{\#\#}P=0.002$, Ad-EGFP vs. Ad-AAA-EGFP). $n = 33-36$, combined data from $n = 3$ independent experiments. Error bars indicate mean \pm SE.

regulation. In addition to updating Fig. 5e,f and Supplementary Fig. 9b,c with this current data set, we added the text in red font, below, to page 10 as follows:

By pre-incubating iBAs expressing Ad-Them1-EGFP with PMA prior to norepinephrine exposure, which would result in the further dissolution of puncta, we observe an increase in OCR values relative to no PMA (Fig. 5d). **The addition of PMA to iBAs led to a progressive increase in values of extracellular acidification rate (ECAR) until the time of NE exposure in the absence of differences in OCR (Supplementary Fig. 9b). This result suggests that PMA stimulates anaerobic glycolysis, as opposed to fatty acid oxidation, which increases OCR following NE exposure. The PMA-induced increases in anaerobic glycolysis led to observed reductions in fatty acid oxidation, as reflected by OCR in response to norepinephrine stimulation, expressed as a % of baseline (Fig. 5e). When the effects of PMA treatment on OCR per se are taken into account, the NE-stimulated suppression of OCR on cells transduced with Ad-AAA was maintained compared to cells transduced with Ad-EGFP or Ad-DDD, (Fig. 5f and Supplementary Fig. 9c). These results argue against downstream regulation of these protein constructs by PMA.**

Minor comments

The sentence on page 7 "Because the canonical PKC binding sequence, R/K-X-S-X-R/K where X at +1 is a hydrophobic residue, is not present within the N-terminal Them1 sequence (Fig. 1c), we used an antibody specific for this sequence in immunoblots of Them1 after stimulation with PMA over time (Supplementary Fig. 7) to explore the hypothesis that PKC phosphorylates Them1" sounds off (why probe for a sequence that is already known not to be present). Could the authors please reword this sentence.

The sentence on page 7 identified by the reviewer was marked in blue font and the changes are marked in red font, which we hope makes the strategy and justification clearer as follows:

The canonical PKC binding sequence, R/K-X-S-X-R/K where X at +1 is a hydrophobic residue, is not present within the N-terminal Them1 sequence (Fig. 1c). However, our pathway

suggested that PKC was involved in the dissolution of puncta. To examine this experimentally, we used an antibody specific for the canonical PKC binding sequence, which showed that no antibody recognition of PKC-mediated S-phosphorylated Them1 occurred (Supplementary Fig. 8). Instead, at least six proteins in iBAs were time-dependently phosphorylated by PKC after PMA stimulation (Supplementary Fig. 8). These results support that Them1 per se is not a PKC substrate but that PKC may be indirectly involved in the dissolution of puncta after PMA stimulation (Fig. 2c).

Reviewer 3

Reviewer #3 (Remarks to the Author):

The authors have performed several new experiments and significantly revised the manuscript to address all of the reviewers' comments and should be commended for improving the study. Attention to the following minor issues is required:

1. The phosphoproteomic data for Them1 supports their argument but is presented as a normalized aggregate abundance of N-terminal phosphopeptides (Fig 1a). Why are the changes at the individual sites, most of which can apparently be measured specifically, not presented? The choice of a phosphopeptide from HSL as a reference for normalization should also be explained – what is the evidence that this site is not affected by PMA?

To address the first concern, the aggregate data in Figure 1a were used because the quantitation was MS1 and we cannot assign the exact location of the phosphorylation because there are several sites possible with a single phosphorylation and each unique isomeric phosphorylated peptide has the exact same mass. The data were collected in a data dependent fashion and this allowed identification of site-specific phosphorylation with MS/MS spectra have by defining ions for the location of the phosphorylation however, the quantitation at the MS1 level cannot be assigned to specific sites as the peptides are not sufficiently resolved in the chromatographic dimension and the masses are identical. Thus, the exact location/locations of the phosphorylation are ambiguous. With this in mind, we labeled phosphorylation events as the aggregate abundance of N-terminal phosphopeptides.

For the second concern, HSL was chosen as a housekeeping phosphopeptide because it was observed in all samples and was not significantly altered due to treatment (FC=# p-value##). The average fold change was 0.92 with a CV of 16.3% and a *p* value > 0.5, indicating no change due to treatment and a relatively low %CV across time points, making it a good internal standard similar to the use of GAPDH or actin in traditional immunoblots. The data for this were included in the revised manuscript as Supplementary Fig. 3 and in the figure legend to Fig. 1a was also revised on page 35.

2. Typo: phosphorylated peptide RSS(phospho)QGVLHMPLYTSPIVK....

The typo was corrected on page 20 of the revised manuscript.

3. The scheme presented in Fig.1f needs further modification. DAG released by ATGL is in the sn-1,3 or sn-2,3 stereochemical isoforms, so that TAG-derived DAG cannot activate protein kinase C (which requires sn-1,2 DAG). It is possible that some DAG released in this way may

undergo isomerization, but this should be discussed.

Eichmann et al., (2012). Studies on the substrate and stereo/regioselectivity of adipose triglyceride lipase, hormone-sensitive lipase, and diacylglycerol-O-acyltransferases. *J Biol Chem* 287, 41446-41457.

Thank you for this important suggested change. We revised Fig. 1f with a dashed line to indicate that the stereo isoform would be required, added this information to the text, page 6 paragraph 2, added the reference (reference 14), and changed the legend to Fig. 2 accordingly on page 35.